# Exponential-Wrapped Mechanisms: Differential Privacy on Hadamard Manifolds Made Practical

**Yangdi Jiang**[1], **Xiaotian Chang**[1], **Lei Ding**[2], **Linglong Kong**[3] , **Bei Jiang**[3]
[1] NTU Singapore, {`yangdi.jiang, xiaotian.chang`}`@ntu.edu.sg`
[2] University of Manitoba, `lei.ding@umanitoba.ca`
[3] University of Alberta, {`lkong, bei1`}`@ualberta.ca`

## Abstract

We propose a general and computationally efficient framework for achieving differential privacy (DP) on Hadamard manifolds, which are complete and simply connected Riemannian manifolds with non-positive curvature. Leveraging the Cartan-Hadamard theorem, we introduce Exponential-Wrapped Laplace and Gaussian mechanisms that achieve $\varepsilon$-DP, $(\varepsilon, \delta)$-DP, Gaussian DP (GDP), and Rényi DP (RDP) without relying on computationally intensive MCMC sampling. Our methods operate entirely within the intrinsic geometry of the manifold, ensuring both theoretical soundness and practical scalability. We derive utility bounds for privatized Fréchet means and demonstrate superior utility and runtime performances on both synthetic data and real-world data in the space of symmetric positive definite matrices (SPDM) and hyperbolic space. To our knowledge, this work constitutes the first unified extension of multiple DP notions to general Hadamard manifolds with practical and scalable implementations.

## 1 Introduction

Recent advances in AI and machine learning have spurred interest in analyzing complex data types, particularly those residing on nonlinear manifolds. Among these, Hadamard manifolds, such as hyperbolic space and the space of symmetric positive definite matrices (SPDM), play pivotal roles. Hyperbolic spaces provide efficient representations for hierarchical structures via hyperbolic embeddings (Nickel and Kiela, 2017; Cetin et al., 2023), enhancing both performance and interpretability in odels for tree-structured data (Sarkar, 2011; Chamberlain et al., 2017; Ganea et al., 2018; Peng et al., 2021). SPDM spaces are critical in medical imaging, especially for modelling water diffusion in diffusion tensor imaging (Basser et al., 1994; Le Bihan et al., 2001), and have found utility in shape analysis and computer vision tasks such as segmentation and motion tracking (Fillard et al., 2005; 2007; Medioni et al., 2000; Brox et al., 2006; Weickert and Brox, 2002; Weickert and Hagen, 2005). The rising importance of manifold-structured data, particularly in the biomedical domains, naturally raises privacy concerns, necessitating tailored privacy mechanisms that respect the underlying geometry.

Differential privacy (DP) (Dwork et al., 2006b) offers a rigorous mathematical framework for quantifying and preserving privacy. While many mechanisms have been developed for Euclidean data (McSherry and Talwar, 2007; Barak et al., 2007; Wasserman and Zhou, 2010; Reimherr and Awan, 2019), they often perform poorly on manifold-valued data due to geometric incompatibility. These traditional methods typically operate extrinsically by embedding manifold data into Euclidean space, which can distort geometric structure and result in substantial utility loss. As demonstrated in Reimherr et al. (2021), respecting the intrinsic geometry of the data leads to mechanisms that provide significantly better trade-offs between privacy and utility. This observation highlights the need for privacy-preserving methods that are not only theoretically sound but also tailored to the geometric nature of nonlinear data through tools from Riemannian geometry.

The differential privacy framework was first extended to general Riemannian manifolds by Reimherr et al. (2021), who introduced the Riemannian Laplace mechanism to achieve $\varepsilon$-DP. Since then,

various mechanisms have been proposed to ensure $\varepsilon$-DP on manifolds (Soto et al., 2022; He et al., 2025), with applications in mobile crowd sensing (Li et al., 2024) and federated learning (Huang et al., 2024). Han et al. (2024) further developed a differentially private Riemannian optimization framework by perturbing the Riemannian gradient in the tangent space. However, extensions of other privacy notions such as $(\varepsilon, \delta)$-DP and Gaussian DP (GDP) remain limited. For example, Utpala et al. (2023b) extended $(\varepsilon, \delta)$-DP only to a single manifold, the SPDM space equipped with the Log-Euclidean metric, which renders the space flat (Arsigny et al., 2007) and enables closed-form computations. However, this choice limits generality, as the framework does not extend to other manifolds, nor to the SPDM space under either the more geometrically faithful affine-invariant metric or the computationally faster and numerically more stable Log-Cholesky metric. Similarly, Jiang et al. (2023) extended GDP to general manifolds, but their calibration algorithm is restricted to constant-curvature spaces and entails high computational cost. Notably, the sampling procedures required by both the Riemannian Laplace and Gaussian mechanisms (Reimherr et al., 2021; Jiang et al., 2023) depend on Markov Chain Monte Carlo (MCMC), which becomes computationally expensive in high-dimensional or geometrically complex spaces such as SPDM. These challenges underscore the need for broader extensions of differential privacy frameworks to general manifolds and the development of more computationally efficient mechanisms suited for practical applications.

We summarize our key contributions below:

- **Unified Extension of DP Notions**: We introduce the first mechanisms to extend $(\varepsilon, \delta)$-DP, Gaussian DP (GDP), and Rényi DP (RDP) to general Hadamard manifolds. Notably, this includes the first RDP mechanism applicable beyond Euclidean spaces.

- **Efficient and Scalable Implementation**: Our proposed Exponential-Wrapped mechanisms avoid computationally intensive MCMC procedures and instead rely on simple sampling from tangent space distributions followed by the exponential map, enabling efficient and scalable deployment.

- **Strong Empirical Performance**: Through comprehensive simulations on SPDM manifolds under multiple metrics, as well as on hyperbolic space, together with real-data experiments on SPDM manifolds, we demonstrate that the proposed mechanisms consistently surpass traditional Riemannian approaches in utility while markedly reducing computational runtime. Notably, the Exponential-Wrapped Gaussian mechanism achieves substantial utility improvements in high-dimensional regimes with stringent privacy budgets.

This paper is organized as follows. Appendix B reviews key concepts from Riemannian geometry (Lee, 2006; Petersen, 2006; Pennec et al., 2019; Said, 2021; Grigoryan, 2009) and differential privacy (Dwork and Roth, 2014; Mironov, 2017; Dong et al., 2021; 2022). Section 2 introduces the Exponential-Wrapped distribution and its calibration for achieving $(\varepsilon, \delta)$-DP, GDP, and RDP. Section 3.1 addresses the release of differentially private Fréchet means and establishes theoretical utility guarantees for our mechanisms. Section 4 presents numerical simulations and real-world experiments, with additional details provided in Appendix 4.2. All proofs are given in Appendix A.

## 2 DIFFERENTIAL PRIVACY ON HADAMARD RIEMANNIAN MANIFOLDS

### 2.1 EXPONENTIAL-WRAPPED DISTRIBUTION

In measure-theoretic terms, the Exponential-Wrapped Probability is the push-forward of the tangent space probability via the exponential map. For a manifold $\mathcal{M}$ with dimension $d > 1$, wrapping a density around the manifold involves volume distortion. This occurs because the exponential map typically does not preserve the area between the Lebesgue measure on the tangent space and the reference measure on the manifold.

Let $\mathcal{M}$ be a manifold with the Riemannian volume measure $\nu$. Given $\mu$, a probability distribution on $T_p\mathcal{M}$ with a probability density $h$ w.r.t the Lebesgue measure $\lambda_p$ on $T_p\mathcal{M}$, the corresponding Exponential-Wrapped distribution is defined as the push-forward of $\mu$ by the exponential, $\Lambda = \mathrm{Exp}_{p*}\mu$, where the $*$ refers to the push-forward by $\mathrm{Exp}_p$, such that $\Lambda(A) = \mu\left(\mathrm{Log}_p(A)\right)$ for any $A$ in the Borel $\sigma$-algebra of $\mathcal{M}$. Since we assume $\mathcal{M}$ is a Hadamard manifold, $\mathrm{Log}_p$ is defined everywhere on $\mathcal{M}$ for any $p \in \mathcal{M}$. If follows that the density $g$ of $\Lambda$ can be expressed from $h$ and a

volume change term,

$$g(q) = \frac{d\Lambda}{d\nu}(q) = \frac{d\,\mathrm{Exp}_{p*}(\lambda_p)}{d\nu} \frac{d\Lambda}{d\,\mathrm{Exp}_{p*}(\lambda_p)}(q) = \frac{d\,\mathrm{Exp}_{p*}(\lambda_p)}{d\nu}(q) h(\mathrm{Log}_p q) = \frac{h(\mathrm{Log}_p q)}{J_p(\mathrm{Log}_p(q))},$$

where $J_p(u) := |\det(D_u \mathrm{Exp}_p)|$ with $D_u \mathrm{Exp}_p : T_p\mathcal{M} \to T_{\mathrm{Exp}_p(u)}\mathcal{M}$ denote the differential of $\mathrm{Exp}_p$ at $u \in T_q\mathcal{M}$.

The most attractive property of the Exponential-Wrapped Distribution is its straightforward sampling procedure. In order to sample from $g$, it suffices to sample from $h$: if $U_1, \ldots, U_n$ are i.i.d. random variables on a tangent space $T_p\mathcal{M}$ following the density $h$, then $X_1 = \mathrm{Exp}_p(U_1), \ldots, X_n = \mathrm{Exp}_p(U_n)$ are i.i.d. random variables on $\mathcal{M}$ following the density $g$. For a more detailed discussion on Exponential-Wrapped Distribution, please refer to Chevallier et al. (2022).

## 2.2 Exponential-Wrapped Laplace Mechanism

**Definition 2.1** (Exponential-Wrapped Laplace Distribution). *Let $\mathcal{M}$ be a Hadamard Riemannian manifold with measure $\nu$, we define a probability density function w.r.t $\nu$ as*

$$g(y) \propto \frac{1}{J_{p_0}(\mathrm{Log}_{p_0} y)} \exp\left(-\frac{\|\mathrm{Log}_{p_0} y - \mathrm{Log}_{p_0} \eta\|}{\sigma}\right).$$

*We called this distribution an **Exponential-Wrapped Laplace Distribution** with **footpoint** $p_0$, **center** $\eta$ and **rate** $\sigma > 0$. We denote it as $\mathrm{EWL}(p_0, \eta, \sigma)$.*

The Exponential-Wrapped Laplace Distribution is the push-forward probability of the tangent space probability defined by the probability density $h(u) \propto \exp\{-\|u - \mathrm{Log}_{p_0} \eta\|/\sigma\}$. We present the following theorem to demonstrate how it can be used to achieve $\varepsilon$-DP.

Denote $\mathcal{D} = \{X_1, X_2, \ldots, X_n\}$ as the confidential dataset and $\mathcal{D}' = \{X_1', X_2, \ldots, X_n\}$ its neighbouring dataset, where without loss of generality we assume they differ in the first record.

**Theorem 2.1** (Exponential-Wrapped Laplace Mechanism). *Let $\mathcal{M}$ be a Hadamard Riemannian manifold and $f$ be a $\mathcal{M}$-valued summary with sensitivity $\Delta$.[1] The Exponential-Wrapped Laplace mechanism, which output $Y \sim \mathrm{EWL}(p_0, f(\mathcal{D}), \Delta/\varepsilon)$, satisfies $\varepsilon$-DP.*

Compared to the Riemannian Laplace mechanism proposed by Reimherr et al. (2021), the Exponential-Wrapped Laplace mechanism defined above offers two primary advantages. First, our method only requires a rate of $\Delta/\varepsilon$ to achieve $\varepsilon$-DP across all Hadamard manifolds, homogeneous or not. This is more efficient than the Riemannian Laplace mechanism, which necessitates a rate of $2\Delta/\varepsilon$ for non-homogeneous manifolds. Second, our approach is easier to implement and less computationally complex. The Riemannian Laplace mechanism relies on MCMC sampling, which is computationally intensive due to prolonged burn-in iterations and frequent recalculations of Riemannian distances. These computations escalate in cost with increasing manifold dimensionality. Even in SPDM space with the Rao-Fisher affine invariant metric, where efficient sampling techniques for the Riemannian Laplace Distribution exist (Hajri et al., 2016) – MCMC procedures remain necessary, and the choice of proposal distribution critically affects convergence. In contrast, sampling from the Exponential-Wrapped Laplace Distribution is straightforward: it involves 1) sampling from $u \sim h(u) \propto \exp\{-\|u - \mathrm{Log}_{p_0} \eta\|/\sigma\}$ and 2) computing $\mathrm{Exp}_{p_0} u$. For more details on the sampling procedure, see Appendix C.1.

**Remark 1.** *Note that there is no restriction on the choice of footpoint $p_0$ in the Exponential-Wrapped Laplace mechanism. However, its selection can have an impact on the performance of the mechanism. Furthermore, to be compliant with the differential privacy definition, the selection of the footpoint $p_0$ cannot be based on the private dataset $\mathcal{D}$. For more discussion on the selection of footpoint, see Section 3.2.*

## 2.3 Exponential-Wrapped Gaussian Mechanism

Beyond the Laplace mechanism, the Gaussian mechanism stands as one of the most prevalent tools in DP (Dwork and Roth, 2014; Balle and Wang, 2018a). This section introduces the Exponential-

---

[1]A summary $f$ is said to have a sensitivity of $\Delta < \infty$, with respect to $d(\cdot, \cdot)$, if we have $d(f(\mathcal{D}), f(\mathcal{D}')) \leq \Delta$ for any two datasets $\mathcal{D} \simeq \mathcal{D}'$.

Wrapped Gaussian mechanism, calibrated to achieve $(\varepsilon, \delta)$-DP, RDP, and GDP. Initially, we will define the Exponential-Wrapped Gaussian Distribution as follows.

**Definition 2.2** (Exponential-Wrapped Gaussian Distribution). *Let $\mathcal{M}$ be a Hadamard Riemannian manifold with reference measure denoted by $\nu$, we define a probability density function w.r.t $\nu$ as*

$$g(y) \propto \frac{1}{J_{p_0}(\text{Log}_{p_0} y)} \exp\left( -\frac{\| \text{Log}_{p_0} y - \text{Log}_{p_0} \eta \|^2}{2\sigma^2} \right).$$

*We called this distribution an **Exponential-Wrapped Gaussian Distribution** with footpoint $p_0$, center $\eta$, and rate $\sigma > 0$. We denote it as $\text{EWG}(p_0, \eta, \sigma)$.*

The Exponential-Wrapped Gaussian Distribution is defined as the push-forward of the multivariate Gaussian distribution, characterized by a mean of $\text{Log}_{p_0} \eta$ and a covariance of $\sigma^2 \mathbf{I}$, on the tangent space $T_p \mathcal{M}$. We present the following theorem to demonstrate how it can be used to achieve $(\varepsilon, \delta)$-DP.

**Theorem 2.2.** *Let $\mathcal{M}$ be a Hadamard Riemannian manifold and $f$ be an $\mathcal{M}$-valued summary. The Exponential-Wrapped Gaussian mechanism, which output $Y \sim \text{EWG}(p_0, f(\mathcal{D}), \sigma)$, satisfies $(\varepsilon, \delta)$-DP if and only if the $\sigma$ satisfies following condition,*

$$\Phi\left( -\frac{\sigma\varepsilon}{\Delta_{p_0}} + \frac{\Delta_{p_0}}{2\sigma} \right) - e^\varepsilon \Phi\left( -\frac{\sigma\varepsilon}{\Delta_{p_0}} + \frac{\Delta_{p_0}}{2\sigma} \right) \le \delta, \tag{1}$$

*where $\Delta_{p_0} = \sup_{D \simeq D'} \| \text{Log}_{p_0}(f(\mathcal{D})) - \text{Log}_{p_0}(f(\mathcal{D}')) \|$.*

Theorem 2.2 shares similarities with the analytic Gaussian mechanism in Balle and Wang (2018a). Primary distinction lies in the use of $\Delta_{p_0}$ rather than the standard sensitivity $\Delta$ in inequality (1). This substitution generally does not pose significant challenges; if $\Delta_{p_0}$ proves difficult to compute, $\Delta$ can be used instead in (1) as $\Delta \ge \Delta_{p_0}$ since $\text{Log}_{p_0}$ is a contraction for Hadamard manifolds.

Implementing the Exponential-Wrapped Gaussian mechanism for $(\varepsilon, \delta)$-DP is straightforward. We follow a similar procedure as in Algorithm 1. After determining the appropriate $\sigma$ numerically from inequality (1)—using a method such as that proposed in Balle and Wang (2018a)—one can proceed by first sampling $\mathbf{u}$ from the tangent Gaussian distribution $\mathcal{N}_{\text{tang}}(\mathbf{0}, \sigma^2 \mathbf{I}_d)$. The privatized summary is then computed as $\text{Exp}_{p_0}(\mathbf{u} + \text{Log}_{p_0}(f(\mathcal{D})))$. For more details on the sampling procedure, see Appendix C.2.

Suppose $\mathcal{M}$ is the space of SPDM equipped with Log-Euclidean metric, the Exponential-Wrapped Gaussian mechanism with footpoint $p_0 = \mathbf{I}$ reduces to the tangent Gaussian mechanism in Utpala et al. (2023b). Hence, the Exponential-Wrapped Gaussian mechanism is a generalization of the tangent Gaussian mechanism, as our mechanism can be employed for any Hadamard manifold equipped with any Riemannian metric. This makes our Exponential-Wrapped Gaussian mechanism the first working mechanism to achieve $(\varepsilon, \delta)$-DP in SPDM under the non-Log-Euclidean metric.

Similar to how the Euclidean Gaussian Distribution can be used to achieve GDP, we can calibrate the Exponential-Wrapped Gaussian Distribution to achieve GDP in the following manner.

**Theorem 2.3** (Wrapped Gaussian Mechanism for GDP). *Let $\mathcal{M}$ be a Hadamard Riemannian manifold and $f$ be an $\mathcal{M}$-valued summary with global sensitivity $\Delta$. The Exponential-Wrapped Gaussian mechanism, which outputs $Y \sim \text{EWG}(p_0, f(\mathcal{D}), \Delta/\mu)$, satisfies $\mu$-GDP.*

Previously, Jiang et al. (2023) introduced the Riemannian Gaussian mechanism to achieve $\mu$-GDP. However, our approach presents significant advantages in both calibration and sampling. Firstly, the Riemannian Gaussian mechanism requires the resolution of infinitely many integral inequalities to calibrate the rate $\sigma$ for a given privacy budget $\mu$. The calibration algorithm provided by Jiang et al. (2023) is only applicable to constant curvature spaces and is computationally intensive, involving grid searches and MCMC techniques to compute the integrals. In contrast, our method simplifies calibration to a straightforward calculation: $\sigma = \Delta/\mu$. Secondly, like the Riemannian Laplace distribution, sampling from the Riemannian Gaussian distribution involves complex processes (detailed in section 2.2). Our sampling technique is considerably simpler, requiring only the sampling from a multivariate Gaussian distribution followed by computations using $\text{Exp}_{p_0}$ and $\text{Log}_{p_0}$. The complete algorithm is detailed in Algorithm 1.

In a similar fashion, we can use the Exponential-Wrapped Gaussian Distribution to achieve RDP.

---

**Algorithm 1** Exponential-Wrapped Gaussian Mechanism for $\mu$-GDP

---
**Input:** sensitivity $\Delta$, privacy budget $\mu$, query result $f(\mathcal{D})$, footpoint $p_0$.
**Output:** privatized query result $\tilde{f}(\mathcal{D})$

  1: **Sample** $\mathbf{v} \sim \mathcal{N}_{\text{tang}}(\mathbf{0}, \mathbf{I}_d)$.
  2: **Compute** $\mathbf{u} = \text{Log}_{p_0} f(\mathcal{D}) + \sigma \mathbf{v}$ with $\sigma = \Delta/\mu$ and $\tilde{f}(\mathcal{D}) = \text{Exp}_{p_0} \mathbf{u}$.
  3: **Return**: $\tilde{f}(\mathcal{D})$.

---

**Theorem 2.4** (Wrapped Gaussian Mechanism for Rényi DP). *Let $\mathcal{M}$ be a Hadamard Riemannian manifold and $f$ be an $\mathcal{M}$-valued summary with global sensitivity $\Delta$. The Exponential-Wrapped Gaussian mechanism, which outputs $Y \sim \text{EWG}(p_0, f(\mathcal{D}), \Delta/\sqrt{2\varepsilon/\alpha})$, satisfies $(\alpha, \varepsilon)$-RDP.*

## 3 DIFFERENTIALLY PRIVATE FRÉCHET MEAN AND UTILITY GUARANTEE

### 3.1 DIFFERENTIALLY PRIVATE FRÉCHET MEAN

For a comprehensive overview of the Fréchet mean in the context of DP, please refer to Reimherr et al. (2021). Consider a set of data $x_1, \ldots, x_n$ on the manifold $\mathcal{M}$. The Euclidean sample mean can be generalized to Riemannian manifolds as the sample Fréchet mean, defined as the minimizer of the sum-of-squared distances to the data points, $\bar{x} = \arg\min_{x \in \mathcal{M}} \sum_{i=1}^{n} d(x, x_i)^2$. The properties of Hadamard manifolds guarantee the existence and uniqueness of the Fréchet mean. To ensure the sensitivity of the sample Fréchet mean is finite, we need the following assumption:

**Assumption 1.** *The data $\mathcal{D} \subseteq B_r(m_0)$ for some $m_0 \in \mathcal{M}$, $r < \infty$.*

The assumption that data lies within a geodesic ball is standard in the field of DP and should not raise concerns (Reimherr et al., 2021; Soto et al., 2022). Consider two datasets $\mathcal{D} \simeq \mathcal{D}'$ on $\mathcal{M}$, and denote $\bar{x}$ and $\bar{x}'$ as the two sample Fréchet means of $\mathcal{D}$ and $\mathcal{D}'$ respectively. Under Assumption 1, we have $d(\bar{x}, \bar{x}') \leq 2r/n$.

### 3.2 UTILITY GUARANTEE

We now analyze the expected utility of our mechanisms in terms of the expected Riemannian distance to the sample Fréchet mean $\bar{x}$.

**Theorem 3.1.** *Let $\mathcal{M}$ be a $d$-dimensional Hadamard manifold and assume assumption 1 holds. Denote $\tilde{x}_{\text{EWL}}$ as a sample drawn from an Exponential-Wrapped Laplace Distribution with footpoint $p_0$, center $\bar{x}$ and rate $\sigma = 2r/(n\varepsilon)$. $\tilde{x}_{\text{EWL}}$ is $\varepsilon$-DP and we have*

$$\mathbb{E}\, d(\tilde{x}_{\text{EWL}}, \bar{x}) \leq \sigma d + 2d(p_0, \bar{x}). \tag{2}$$

*Similarly, denote $\tilde{x}_{\text{EWG}}$ as a sample drawn from an Exponential-Wrapped Gaussian Distribution with footpoint $p_0$, center $\bar{x}$ and rate $\sigma = 2r/(n\mu)$. $\tilde{x}_{\text{EWG}}$ is $\mu$-GDP and we have,*

$$
\begin{aligned}
\mathbb{E}\, d(\tilde{x}_{\text{EWG}}, \bar{x}) &\leq \sigma\sqrt{\frac{\pi}{2}} L_{1/2}^{d/2-1}\left(-\frac{d^2(p_0, \bar{x})}{2}\right) + d(p_0, \bar{x}) \\
&\leq \sigma\sqrt{2}\frac{\Gamma((d+1)/2)}{\Gamma(d/2)} + 2d(p_0, \bar{x}),
\end{aligned}
\tag{3}
$$

*where $L_{1/2}$ denote the Laguerre polynomials. If we impose the additional of $\text{Sec}_{\mathcal{M}} > K$ for some $K \leq 0$, we have,*

$$\mathbb{E}\, d(\tilde{x}_{\text{EWL}}, \bar{x}) \leq \frac{\sinh(\sqrt{K}r)}{\sqrt{K}r}\sigma d, \tag{4}$$

$$\mathbb{E}\, d(\tilde{x}_{\text{EWG}}, \bar{x}) \leq \frac{\sinh(\sqrt{K}r)}{\sqrt{K}r}\sigma\sqrt{2}\frac{\Gamma((d+1)/2)}{\Gamma(d/2)}. \tag{5}$$

*where $\Gamma$ denotes the gamma function.*[2]

---
[2]$\text{Sec}_{\mathcal{M}}$ denotes the sectional curvature of $\mathcal{M}$.

Observe that the footpoint $p_0$ appears explicitly in the utility bounds equation 2 and equation 3, highlighting its direct impact on the utility of the mechanisms. To tighten these bounds, it is desirable to minimize the distance $d(p_0, \bar{x})$. When the data are well-dispersed within a ball $B_r(m_0)$, the Fréchet mean typically lies near the center $m_0$, making it a natural choice for the footpoint. However, if prior knowledge suggests that the majority of the data are concentrated in a smaller region $R \subset B_r(m_0)$, selecting the center of $R$ as the footpoint may yield better utility. In the absence of such prior information, it may be beneficial to allocate a portion of the privacy budget to privately estimate a suitable footpoint. See Appendix G for a concrete DP mechanism for selecting a data-dependent footpoint $p_0$.

Furthermore, as the dimension $d$ increases, the leading terms in the utility bounds dominate, reducing the relative influence of the footpoint, provided the data radius $r$ in Assumption 1 remains fixed. This implies that in high-dimensional settings, the influence of the footpoint on utility becomes relatively less significant. Indeed, this trend is reflected in our simulation results on the SPDM space equipped with the affine-invariant metric, as shown in Section 4.

By contrast, the bounds in equation 4 and equation 5 illustrate how the geometry of the underlying manifold influences utility through curvature. The sectional curvature lower bound $K$ acts as a regularizer: smaller values of $K$ correspond to geometries closer to flat and yield tighter utility bounds. In the limiting case of a flat manifold, these bounds simplify to equalities:

$$\mathbb{E}\, d(\tilde{x}_{\mathrm{EWL}}, \bar{x}) = \sigma d, \quad \mathbb{E}\, d(\tilde{x}_{\mathrm{EWG}}, \bar{x}) = \sigma\sqrt{2}\, \frac{\Gamma((d+1)/2)}{\Gamma(d/2)},$$

in which the footpoint no longer affects the utility. This phenomenon is further confirmed in our simulations on SPDM spaces equipped with the Log-Cholesky and Log-Euclidean metrics, as presented in Section 4.

## 4    SIMULATION AND EXPERIMENT

We evaluate the performance of our Exponential-Wrapped mechanisms for releasing GDP-compliant Fréchet means. Experiments are conducted on the manifold of symmetric positive definite matrices (SPDM), a standard space in medical imaging (Pennec et al., 2019; Said et al., 2017; Hajri et al., 2016). Appendix B.3 reviews the geometry of SPDM under three metrics, while Appendix B.4 reviews the geometry of the hyperbolic space. Section 4.1 describes the simulation setup and presents results. Real-world experiments on the OCTMNIST dataset are provided in section 4.2.

### 4.1    NUMERICAL SIMULATION

For the simulation study, we focus on releasing the GDP Fréchet mean on the SPDM spaces $S_m^+$ under three different Riemannian metrics: the Log-Cholesky metric, the Log-Euclidean metric, and the affine-invariant metric[3], and on Hyperbolic space $\mathbb{H}_d$. We compare the performance of our Exponential-Wrapped Gaussian (EWG) mechanism, described in Section 2.3, with the Riemannian Laplace (RL) mechanism proposed in Reimherr et al. (2021). A Riemannian Laplace mechanism that satisfies $\varepsilon$-DP can also be interpreted as satisfying $\mu$-GDP, with the correspondence given by $\varepsilon = \log[(1 - \Phi(-u/2))/\Phi(-u/2)]$ (Liu et al., 2022).

For SPDM spaces $S_m^+$, we generate samples $x_1, \ldots, x_n$ from the geodesic ball $B_r(\mathbf{I}_m)$ using the Wishart distribution as in Reimherr et al. (2021, Supplemental 1.2.1). The Fréchet mean $\bar{x}$ is computed using formulas equation 9 and equation 8 for the Log-Cholesky and Log-Euclidean metrics, respectively, while the gradient descent procedure from Fletcher and Joshi (2004); Reimherr et al. (2021) is used for the affine-invariant metric. To implement the RL mechanism, we follow the method of Reimherr et al. (2021); Hajri et al. (2016), using a burn-in of 10,000 iterations to sample from the Riemannian Laplace distribution. For our EWG mechanism, we use the method described in Algorithm 1 and Appendix C.3, sampling from the Exponential-Wrapped Gaussian distribution centred at the fixed footpoint $p_0 = \mathbf{I}_m$.

Throughout these simulations, we fix the sample size at $n = 40$ and the data radius at $r = 1.5$ to ensure constant sensitivity $\Delta$. With $\Delta$ fixed, we vary the privacy budget $\mu \in \{0.1, 0.2, \ldots, 0.7, 1, 1.5, 2\}$ and

---

[3]Note that the manifold of SPDM is not inherently Hadamard; this property depends on the choice of Riemannian metric.

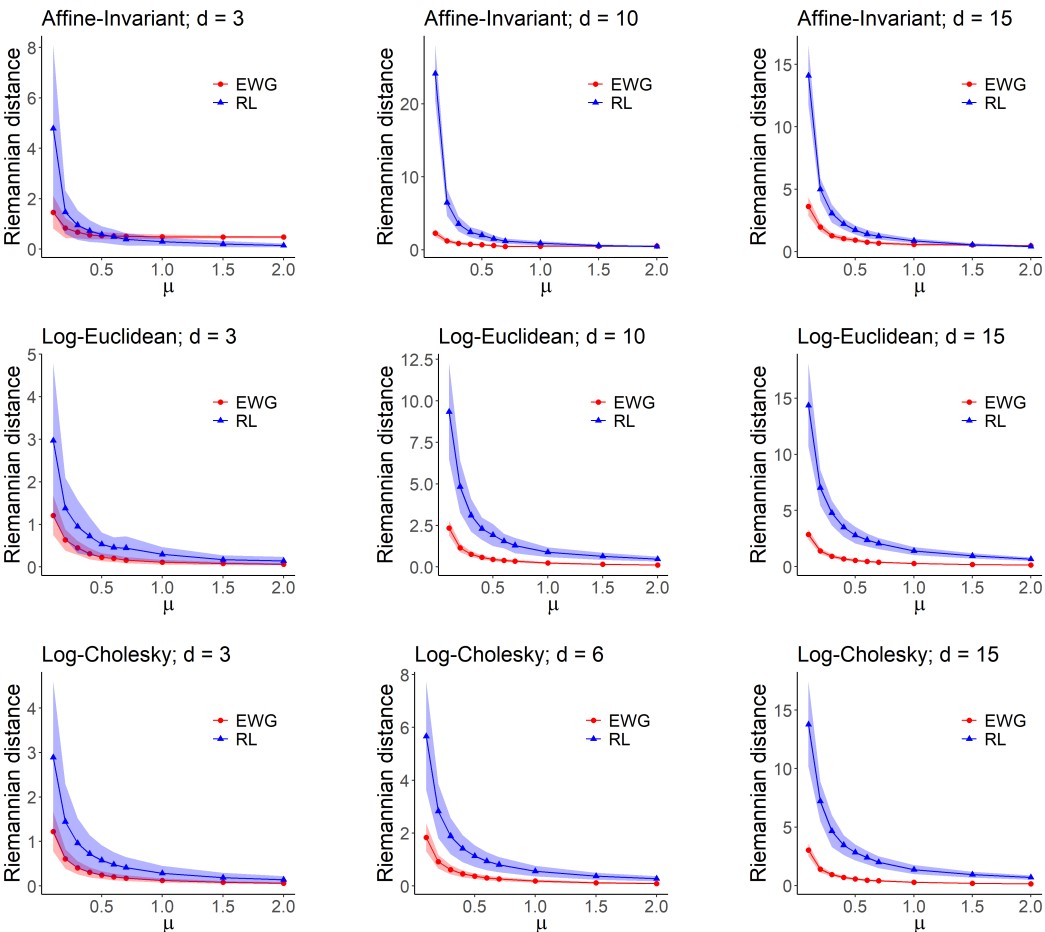

Figure 1: Utility Comparison of EWG and RL Mechanisms across Three Metrics. Blue lines with triangular symbols show the Riemannian distances $d(\bar{x}, \tilde{x}_{\text{RL}}^{\text{gdp}})$, and the red line with circular symbols represent the Riemannian distances $d(\bar{x}, \tilde{x}_{\text{EWG}}^{\text{gdp}})$. Results are shown for the Log-Cholesky (top), Log-Euclidean (middle), and affine-invariant (bottom) metrics.

manifold dimension $d = m(m+1)/2 \in \{3, 10, 15\}$. Let $\tilde{x}_{\text{RL}}^{\text{gdp}}$ and $\tilde{x}_{\text{EWG}}^{\text{gdp}}$ denote the outputs of the RL and EWG mechanisms, respectively. Figure 1 shows the average Riemannian distances $d(\bar{x}, \tilde{x}_{\text{RL}}^{\text{gdp}})$ (blue triangles) and $d(\bar{x}, \tilde{x}_{\text{EWG}}^{\text{gdp}})$ (red circles), computed over 100 independent runs. Shaded regions indicate standard errors around the sample means. Results are organized by metric: the top, middle, and bottom rows correspond to the Log-Cholesky, Log-Euclidean, and affine-invariant metrics, respectively.

Across all three dimensions $d \in \{3, 10, 15\}$ and a wide range of privacy budgets $\mu \in [0.1, 2]$, the EWG mechanism consistently outperforms the RL mechanism under the Log-Cholesky and Log-Euclidean metrics. Under the affine-invariant metric, the EWG mechanism shows degraded performance at moderate to high privacy budgets $\mu \in [0.7, 2]$ when $d = 3$. This degradation reflects the influence of curvature on utility: as the privacy budget increases and less noise is injected, the effect of footpoint misalignment becomes more pronounced. However, for higher dimensions $d = 10, 15$, the EWG mechanism outperforms the RL mechanism across nearly all privacy budgets. In high dimensions, utility is dominated by noise magnitude rather than footpoint alignment, leading to more stable performance.

Similarly, for hyperbolic space $\mathbb{E}_d$, we generate samples $x_1, \ldots, x_n$ uniformly from the geodesic ball $B_r(p)$ with $p = (1, 0, \cdots, 0)$, $r = 1.5$ and $n = 40$ fixed throughout the simulation. Same as the SPDM space equipped with Affine-Invariant metric, the Fréchet mean $\bar{x}$ is computed using the

gradient descent procedure from Fletcher and Joshi (2004); Reimherr et al. (2021). To implement the RL mechanism, we follow the sampling method of Pennec et al. (2019). For our EWG mechanism, we use the method described in Algorithm 1 and Appendix C.4, sampling from the Exponential-Wrapped Gaussian distribution centred at the fixed footpoint $p_0 = (1, 0, \cdots, 0)$. The results, shown in Figure 2, are much more consistent compared to $S_m^+$ with Affine-Invariant metric. Across all dimensions $d \in \{3, 10, 15\}$ and all privacy budget, the EWG mechanism consistently outperforms the RL mechanism, which is similar to the scenarios of $S_m^+$ under Log-Euclidean and Log-Cholesky metrics. This suggests that in spaces with more regular, less exotic geometry, the influence of the footpoint on performance is reduced.

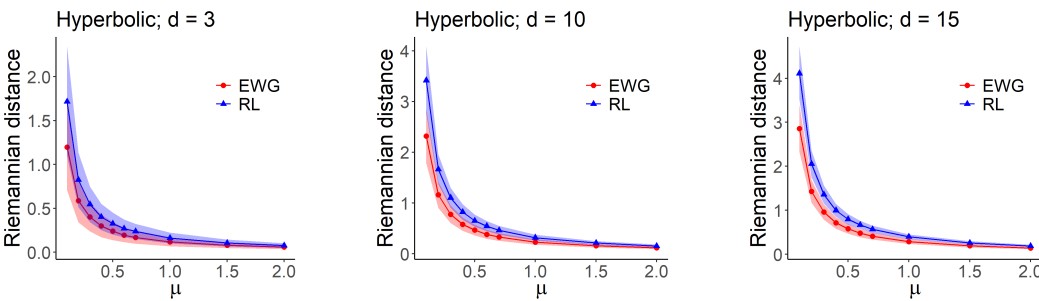

Figure 2: Utility Comparison of EWG and RL Mechanisms on Hyperbolic space $\mathbb{H}_d$. Blue lines with triangular symbols show the Riemannian distances $d(\bar{x}, \tilde{x}_{\text{RL}}^{\text{gdp}})$, and the red line with circular symbols represent the Riemannian distances $d(\bar{x}, \tilde{x}_{\text{EWG}}^{\text{gdp}})$.

In addition to improved utility, the EWG mechanism offers a substantial computational advantage over the RL mechanism. As shown in Table 1, EWG requires significantly less runtime, often by several orders of magnitude, especially as the manifold dimension increases. This efficiency stems from its straightforward sampling procedure (see Section 2.3 and Algorithm 1), in contrast to the RL mechanism's reliance on MCMC with a long burn-in, making EWG more scalable and practical for high-dimensional settings.

## 4.2 REAL-WORLD EXPERIMENT ON OCTMINST DATASET

In this section, we compare our EWG mechanism to the Riemannian Laplace mechanism for releasing the GDP Fréchet mean on the real-world dataset OCTMINST. As one of the 12 standardized 2D datasets in the MedMNIST collection (Yang et al., 2023), OCTMINST consists of $28 \times 28$ greyscale images and is derived from Optical Coherence Tomography (OCT) medical imaging data. Following prior work (Utpala et al., 2023a; Tuzel et al., 2006), we extract covariance descriptors from each image to represent them as points on the space of $5 \times 5$ SPDM, $S_5^+$, equipped with the Log-Euclidean metric. These descriptors serve as structured, manifold-valued features for comparison. The detailed covariance descriptor construction is given below.

To perform tasks such as classification on medical imaging datasets, it is common practice to extract a covariance descriptor from each image, using it as a representative feature of the image. Here, we follow a similar approach as in Utpala et al. (2023a); Tuzel et al. (2006) to extract the covariance descriptors. Let $\mathcal{I} \in \mathbb{R}^{h \times w}$ be an greyscale image of height $h$ and width $w$ and $\mathcal{I}(\mathbf{x})$ denote the pixel intensity at position $x$ and $y$ with $\mathbf{x} = (x, y)$. The covariance descriptor is given by

$$\text{R}_\eta(\mathcal{I}) = \left[ \frac{1}{|\mathcal{S}|} \sum_{\mathbf{x} \in S} (\phi(\mathcal{I})(\mathbf{x}) - \mu)(\phi(\mathcal{I})(\mathbf{x}) - \mu)^T \right] + \eta \mathbf{I},$$

where $\phi(\mathcal{I})(\mathbf{x})$ is defined as the following,

$$\left[ \mathcal{I}(\mathbf{x}), \left| \frac{\partial \mathcal{I}(\mathbf{x})}{\partial x} \right|, \left| \frac{\partial \mathcal{I}(\mathbf{x})}{\partial y} \right|, \left| \frac{\partial^2 \mathcal{I}(\mathbf{x})}{\partial x^2} \right|, \left| \frac{\partial^2 \mathcal{I}(\mathbf{x})}{\partial y^2} \right| \right].$$

We set $\eta = 10^{-6}$ to ensure the covariance descriptors are positive definite. It follows that each covariance descriptor $R_\eta(\mathcal{I})$ is an element of $S_5^+$ with $d = 5(5 + 1)/2 = 15$.

Equipping $S_5^+$ with the Log-Euclidean metric, we have

$$d(R_\eta(\mathcal{I}), \mathbf{I}) \leq \sqrt{5} \max\left\{ |\log(\eta)|, |\log(5 \cdot 255^2 + \eta)| \right\} \tag{6}$$

by following a similar computation as in Utpala et al. (2023a). Note that, different from the experiment in Utpala et al. (2023a), we did not normalize the pixel value/intensity $\mathcal{I}$ to be between $0$ and $1$. Based on (6), the data must reside in $B_r(\mathbf{I})$ where $r$ is determined by the righthand side of (6) and thus the sensitivity for computing Fréchet mean is then $\Delta = \sup_{\bar{x} \simeq \bar{x}'} d(\bar{x}, \bar{x}') \leq 2r/n$.

We compute a covariance descriptor for each image in the dataset, then calculate the sample Fréchet mean $\bar{x}$ under the Log-Euclidean metric. We then release a GDP version $\tilde{x}^{\text{gdp}}$ using both the Exponential-Wrapped Gaussian (EWG) mechanism and the Riemannian Laplace (RL) mechanism (Reimherr et al., 2021), each calibrated to the same privacy budget $\mu \in \{0.1, 0.2, \dots, 0.7, 1, 1.5, 2\}$. For EWG, we fix the footpoint at $p_0 = I_5$ and apply Algorithm 1. For RL, we follow the MCMC-based sampling procedure with 10,000 burn-in steps.

There are four different classes in the OCTMNIST dataset, labelled from $0$ to $3$. We compare the utility between our EWG mechanism and RL mechanism in each of the four classes. Denote $\tilde{x}_{\text{RL}}^{\text{gdp}}, \tilde{x}_{\text{EWG}}^{\text{gdp}}$ as the output of the Riemannian Laplace mechanism and Exponential-Wrapped Laplace mechanism respectively, the plots in Figure 3 display the average Riemannian distances $d(\bar{x}, \bar{x}_{\text{EWG}}^{\text{gdp}})$ (in red with circular symbols) and $d(\bar{x}, \bar{x}_{\text{RL}}^{\text{gdp}})$ (in blue with triangular symbols) across 100 Monte Carlo replications for each class. Shaded regions indicate standard errors around the sample means. Similarly to the numerical simulation, our EWG mechanism achieves better utility across different privacy budgets. These results confirm that EWG provides practical scalability and strong utility guarantees for differentially private inference on real-world manifold-valued data.

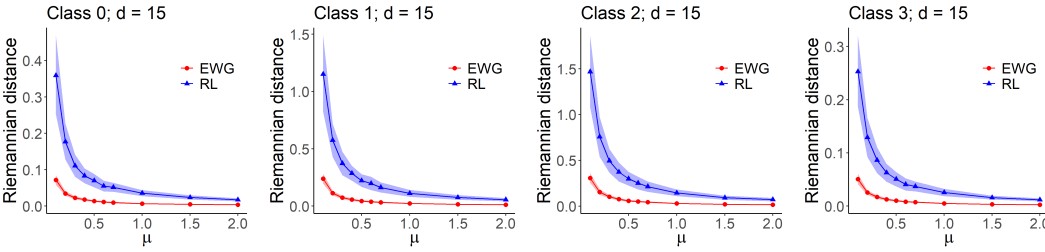

Figure 3: Utility Comparison of EWG and RL Mechanisms for Class 0 to 3 in the `OCTMNIST` data under Log-Euclidean metric. Blue lines with triangular symbols show the Riemannian distances $d(\bar{x}, \tilde{x}_{\text{RL}}^{\text{gdp}})$, and the red line with circular symbols represent the Riemannian distances $d(\bar{x}, \tilde{x}_{\text{EWG}}^{\text{gdp}})$.

## 5 CONCLUSION AND FUTURE DIRECTIONS

We introduced Exponential-Wrapped Laplace and Gaussian mechanisms for achieving differential privacy on Hadamard manifolds. These mechanisms support multiple privacy notions—$(\varepsilon, \delta)$-DP, Rényi DP, and Gaussian DP—and operate entirely within the intrinsic geometry of the manifold. Crucially, they avoid MCMC sampling by leveraging efficient push-forward sampling via the exponential map. Theoretically, we derived utility bounds for both mechanisms that capture the impact of curvature, dimension, and footpoint alignment on the privatized Fréchet mean. Empirically, we showed that the Exponential-Wrapped Gaussian mechanism consistently outperforms the Riemannian Laplace mechanism on flat manifolds and performs competitively on curved manifolds, with substantial improvements in runtime.

Several avenues for future work remain. First, determining an optimal or data-adaptive choice of footpoint $p_0$ could improve utility in non-constant curvature settings. In particular, strategies for privately selecting $p_0$ are worth exploring. Second, extending our framework to manifolds with non-negative curvature, such as spheres, is a natural direction. Finally, we aim to extend our approach beyond Fréchet mean estimation to more complex tasks such as principal geodesic analysis (Huckemann et al., 2010; Fletcher et al., 2003; Zhang and Fletcher, 2013) and regression on manifolds (Cheng and Wu, 2013).

ACKNOWLEDGMENTS

This work was initiated during the first author's PhD studies at the University of Alberta and further revised/finalized during a postdoctoral appointment at Nanyang Technological University. The first author gratefully acknowledges support from the Alberta Innovates scholarship.

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

## A  PROOFS

### A.1  PROOF OF THEOREM 2.1

*Proof.* Denote the Exponential-Wrapped Laplace mechanism as $M$ and its density as $g_1$ corresponding to $f(\mathcal{D})$ and $g_2$ corresponding to $f(\mathcal{D}')$. To show $\mathbb{P}(M(\mathcal{D}) \in S) \leq e^\varepsilon \mathbb{P}(M(\mathcal{D}') \in S)$ for all measurable set $S$, it's sufficient to show that,

$$\frac{g_1(y)}{g_2(y)} \leq e^\varepsilon.$$

Denote $\eta_1 = f(\mathcal{D})$ and $\eta_2 = f(\mathcal{D}')$, we simplify the ratio on the left-hand side,

$$
\begin{aligned}
\frac{g_1(y)}{g_2(y)} &= \frac{\frac{1}{J_{p_0}(\mathrm{Log}_{p_0}(y))} \exp\left(-\frac{\|\mathrm{Log}_{p_0}(y)-\mathrm{Log}_{p_0}(\eta_1)\|}{\sigma}\right)}{\frac{1}{J_{p_0}(\mathrm{Log}_{p_0}(y))} \exp\left(-\frac{\|\mathrm{Log}_{p_0}(y)-\mathrm{Log}_{p_0}(\eta_2)\|}{\sigma}\right)} \\
&= \exp\left\{\frac{1}{\sigma}\left[\|\mathrm{Log}_{p_0}(y) - \mathrm{Log}_{p_0}(\eta_2)\| - \|\mathrm{Log}_{p_0}(y) - \mathrm{Log}_{p_0}(\eta_1)\|\right]\right\} \\
&\leq \exp\left\{\frac{1}{\sigma}\|\mathrm{Log}_{p_0}(\eta_1) - \mathrm{Log}_{p_0}(\eta_2)\|\right\}, \quad \text{triangle inequality} \\
&\leq \exp\left\{\frac{1}{\sigma}d(\eta_1,\eta_2)\right\}, \quad \mathrm{Log}_y \text{ is a contraction for Hadamard manifold} \\
&\leq \exp\left\{\frac{\Delta}{\sigma}\right\} \\
&\leq e^\varepsilon, \quad \text{for } \sigma = \frac{\Delta}{\varepsilon}.
\end{aligned}
$$

$\square$

## A.2 Proof of Theorem 2.2

*Proof.* Let $g_{p_0,\eta,\sigma}$ denote the Exponential-Wrapped Gaussian Distribution with footpoint $p_0$, center $\eta$ and rate $\sigma$. From Balle and Wang (2018a), our Exponential-Wrapped Gaussain mechanism satisfies $(\varepsilon,\delta)$-DP if and only if,

$$
\sup_{\mathcal{D}\simeq\mathcal{D}'} \int_A g_{p_0,\eta_1,\sigma}(y)\,\mathrm{d}\nu(y) - e^\varepsilon \int_A g_{p_0,\eta_2,\sigma}(y)\,\mathrm{d}\nu(y) \leq \delta,
$$

where $A = \{y \mid g_{p_0,\eta_1,\sigma}(y)/g_{p_0,\eta_2,\sigma}(y) \geq e^\varepsilon\}$, $\eta_1 = f(\mathcal{D})$ and $\eta_2 = f(\mathcal{D}')$. We have

$$
\begin{aligned}
&\frac{g_{p_0,\eta_1,\sigma}(y)}{g_{p_0,\eta_2,\sigma}(y)} \\
&= \exp\left\{\frac{1}{2\sigma^2}\left[\|\mathrm{Log}_{p_0}(y) - \mathrm{Log}_{p_0}(\eta_2)\|^2 - \|\mathrm{Log}_{p_0}(y) - \mathrm{Log}_{p_0}(\eta_1)\|^2\right]\right\} \\
&= \exp\left\{\frac{1}{2\sigma^2}\left[-2\left\langle\mathrm{Log}_{p_0}(y) - \mathrm{Log}_{p_0}(\eta_1), \mathrm{Log}_{p_0}(\eta_2) - \mathrm{Log}_{p_0}(\eta_1)\right\rangle + \|\mathrm{Log}_{p_0}(\eta_2) - \mathrm{Log}_{p_0}(\eta_1)\|^2\right]\right\}.
\end{aligned}
$$

Denote $\Delta_{p_0,\eta_1,\eta_2} = \|\mathrm{Log}_{p_0}(\eta_2) - \mathrm{Log}_{p_0}(\eta_1)\|$. It follows that,

$$
A = \left\{y \mid \left\langle\mathrm{Log}_{p_0}(y) - \mathrm{Log}_{p_0}(\eta_1), \mathrm{Log}_{p_0}(\eta_2) - \mathrm{Log}_{p_0}(\eta_1)\right\rangle \leq -\sigma^2\varepsilon + \frac{\Delta_{p_0,\eta_1,\eta_2}^2}{2}\right\}
$$

Apply change of variable with $u = \mathrm{Log}_{p_0} y$, we have

$$
\sup_{\mathcal{D}\simeq\mathcal{D}'} \int_{A^*} \mathcal{N}(u \mid \mathrm{Log}_{p_0}(\eta_1), \sigma^2\mathbf{I})\,\mathrm{d}\lambda(u) - e^\varepsilon \int_{A^*} \mathcal{N}(u \mid \mathrm{Log}_{p_0}(\eta_2), \sigma^2\mathbf{I})\,\mathrm{d}\lambda(u) \leq \delta,
$$

where $\lambda$ is the Lebegue measure on the tangent space $T_{p_0}\mathcal{M}$ and

$$
A^* = \left\{u \mid \left\langle u - \mathrm{Log}_{p_0}(\eta_1), \mathrm{Log}_{p_0}(\eta_2) - \mathrm{Log}_{p_0}(\eta_1)\right\rangle \geq -\sigma^2\varepsilon + \frac{\Delta_{p_0,\eta_1,\eta_2}^2}{2}\right\}.
$$

It follows that,

$$
\int_{A^*} \mathcal{N}(u \mid \mathrm{Log}_{p_0}(\eta_1), \sigma^2\mathbf{I})\,\mathrm{d}\lambda(u) = \Phi\left(-\frac{\sigma\varepsilon}{\Delta_{p_0,\eta_1,\eta_2}} + \frac{\Delta_{p_0,\eta_1,\eta_2}}{2\varepsilon}\right).
$$

Take a similar approach for the second integral, we have

$$
\int_{A^*} \mathcal{N}(u \mid \mathrm{Log}_{p_0}(\eta_2), \sigma^2\mathbf{I})\,\mathrm{d}\lambda(u) = \Phi\left(-\frac{\sigma\varepsilon}{\Delta_{p_0,\eta_1,\eta_2}} - \frac{\Delta_{p_0,\eta_1,\eta_2}}{2\varepsilon}\right).
$$

Finally, we have

$$\Phi\left(-\frac{\sigma\varepsilon}{\Delta_{p_0}} + \frac{\Delta_{p_0}}{2\sigma}\right) - e^\varepsilon \Phi\left(-\frac{\sigma\varepsilon}{\Delta_{p_0}} + \frac{\Delta_{p_0}}{2\sigma}\right) \le \delta,$$

where $\Delta_{p_0} = \sup_{\mathcal{D}\simeq\mathcal{D}'} \Delta_{p_0,\eta_1,\eta_2}$ as needed. $\qquad\square$

### A.3  PROOF OF THEOREM 2.3

*Proof.* Using definition B.3, we need to show the following,

$$\forall\varepsilon \ge 0,\ \sup_{\mathcal{D}\simeq\mathcal{D}'} \int_A g_{p_0,\eta_1,\sigma}(y)\,\mathrm{d}\nu(y) - e^\varepsilon \int_A g_{p_0,\eta_2,\sigma}(y)\,\mathrm{d}\nu(y) \le \delta_\mu(\varepsilon) \qquad (7)$$

where $g$ denotes the density of the Exponential-Wrapped Gaussian Distribution. From the proof in A.2, we have

$$\sup_{\mathcal{D}\simeq\mathcal{D}'} \int_A g_{p_0,\eta_1,\sigma}(y)\,\mathrm{d}\nu(y) - e^\varepsilon \int_A g_{p_0,\eta_2,\sigma}(y)\,\mathrm{d}\nu(y)$$
$$= \Phi\left(-\frac{\sigma\varepsilon}{\Delta_{p_0}} + \frac{\Delta_{p_0}}{2\sigma}\right) - e^\varepsilon \Phi\left(-\frac{\sigma\varepsilon}{\Delta_{p_0}} + \frac{\Delta_{p_0}}{2\sigma}\right).$$

Thus, the equality in (7) holds if and only if $\sigma = \Delta_{p_0}/\mu$. Since $\mathrm{Log}_{p_0}$ is a contraction for any $p_0 \in \mathcal{M}$ (for Hadamard manifold $\mathcal{M}$), we have $\Delta \ge \Delta_{p_0}$ and $\sigma = \Delta/\mu$ achieves $\mu$-GDP as well. $\qquad\square$

### A.4  PROOF OF THEOREM 2.4

*Proof.* Let $M$ denote the Exponential-Wrapped Gaussian mechanism, we have

$$D_\alpha(M(\mathcal{D})\|M(\mathcal{D}'))$$
$$= \frac{1}{\alpha-1} \log \int \frac{1}{J_{p_0}(\mathrm{Log}_{p_0}(y))} \frac{1}{(\sqrt{2\pi}\sigma)^d}$$
$$\exp\left\{-\frac{\alpha}{2\sigma^2}\left[\|\mathrm{Log}_{p_0} y - \mathrm{Log}_{p_0} \eta_1\|^2\right] - \frac{1-\alpha}{2\sigma^2}\|\mathrm{Log}_{p_0} y - \mathrm{Log}_{p_0} \eta_2\|^2\right\}\,d\nu(y)$$
$$= \frac{1}{\alpha-1} \log \exp\left\{-\frac{\alpha(1-\alpha)}{2\sigma^2}\|\mathrm{Log}_{p_0} \eta_1 - \mathrm{Log}_{p_0} \eta_2\|^2\right\},\ \text{completing the squares}$$
$$= \frac{\alpha}{2\sigma^2}\|\mathrm{Log}_{p_0} \eta_1 - \mathrm{Log}_{p_0} \eta_2\|^2$$
$$\le \frac{\alpha}{2\sigma^2} d(\eta_1 - \eta_2)^2,\ \mathrm{Log}_{p_0}\ \text{is a contraction for Hadamard manifolds}$$
$$\le \frac{\alpha}{2\sigma^2}\Delta^2$$
$$\le \varepsilon,\ \text{for } \sigma = \frac{\Delta}{\sqrt{2\varepsilon/\alpha}}.$$

$\qquad\square$

### A.5  PROOF OF THEOREM 3.1

First, we will show the proof for bounds in (2) and (3).

**Lemma 1.** *Let $\mathcal{M}$ be a $d$-dimensional Hadamard manifold.*

1. *Denote $y$ as a sample drawn from an Exponential-Wrapped Laplace Distribution with footpoint $p_0$, center $\eta$ and rate $\sigma$, then we have,*

$$\mathbb{E}\, d(y, \eta) \le \sigma d + 2\|\mathrm{Log}_{p_0} \eta\|.$$

2. *Denote $y$ as a sample drawn from an Exponential-Wrapped Gaussian Distribution with footpoint $p_0$, center $\eta$ and rate $\sigma$, then we have*

$$\mathbb{E}\,d(y,\eta) \leq \sigma\sqrt{\frac{\pi}{2}}L_{1/2}^{d/2-1}\left(-\frac{d(p_0,\eta)^2}{2}\right) + d(p_0,\eta) \leq \sigma\sqrt{2}\frac{\Gamma((d+1)/2)}{\Gamma(d/2)} + 2\|\operatorname{Log}_{p_0}\eta\|.$$

*Proof.* For Exponential-Wrapped Laplace Distribution, denote

$$C(\sigma) = \int \exp\left(-\frac{\|x\|}{\sigma}\right)\,\mathrm{d}\lambda(x),$$

then we have

$$\mathbb{E}d(y,\eta)$$
$$= \int d(y,\eta)\frac{C(\sigma)^{-1}}{J_{p_0}(\operatorname{Log}_{p_0}y)}\exp\left(-\frac{\|\operatorname{Log}_{p_0}y - \operatorname{Log}_{p_0}\eta\|}{\sigma}\right)\,\mathrm{d}\nu(y)$$
$$\leq \int d(y,p_0)\frac{C(\sigma)^{-1}}{J_{p_0}(\operatorname{Log}_{p_0}y)}\exp\left(-\frac{\|\operatorname{Log}_{p_0}y - \operatorname{Log}_{p_0}\eta\|}{\sigma}\right)\,\mathrm{d}\nu(y) + d(p_0,\eta), \text{ triangle inequality}$$
$$= \int \|\operatorname{Log}_{p_0}y\|\frac{C(\sigma)^{-1}}{J_{p_0}(\operatorname{Log}_{p_0}y)}\exp\left(-\frac{\|\operatorname{Log}_{p_0}y - \operatorname{Log}_{p_0}\eta\|}{\sigma}\right)\,\mathrm{d}\nu(y) + d(p_0,\eta)$$
$$= \int \frac{\|u + \operatorname{Log}_{p_0}\eta\|}{C(\sigma)}\exp\left(-\frac{\|u\|}{\sigma}\right)\,\mathrm{d}\lambda(u) + d(p_0,\eta), \ u = \operatorname{Log}_{p_0}y - \operatorname{Log}_{p_0}\eta$$
$$\leq \frac{1}{C(\sigma)}\int \|u\|\exp\left(-\frac{\|u\|}{\sigma}\right)\,\mathrm{d}\lambda(u) + 2d(p_0,\eta), \text{ triangle inequality}$$
$$= \left(\sigma\int_0^\infty r^{d-1}\exp(-r)\,\mathrm{d}r\right)^{-1}\int_0^\infty \sigma^2 r^d\exp(-r)\,\mathrm{d}r + 2d(p_0,\eta), \text{ spherical coordinates}$$
$$= \sigma d + 2d(p_0,\eta).$$

Similarly, for the Exponential-Wrapped Gaussian distribution, we have,

$$\mathbb{E}d(y,\eta)$$
$$= \int d(y,\eta)\frac{(\sqrt{2\pi}\sigma)^{-d}}{J_{p_0}(\operatorname{Log}_{p_0}y)}\exp\left(-\frac{\|\operatorname{Log}_{p_0}y - \operatorname{Log}_{p_0}\eta\|^2}{2\sigma^2}\right)\,\mathrm{d}\nu(y)$$
$$\leq \mathbb{E}d(y,p_0) + d(p_0,\eta).$$

Note that since $(\operatorname{Log}_{p_0}y)/\sigma \sim \mathcal{N}(\operatorname{Log}_{p_0}\eta, \mathbf{I})$, $d(y,p_0)/\sigma$ follows a noncentral chi distribution and have a mean of

$$\sqrt{\frac{\pi}{2}}L_{1/2}^{d/2-1}\left(-\frac{d(p_0,\eta)^2}{2}\right),$$

where $L_{1/2}$ denote the Laguerre polynomials. Thus, we have

$$\mathbb{E}d(y,\eta)$$
$$\leq \sigma\sqrt{\frac{\pi}{2}}L_{1/2}^{d/2-1}\left(-\frac{d(p_0,\eta)^2}{2}\right) + d(p_0,\eta).$$

However, this upper bound is hard to interpret. We will also derive a less tight upper bound but with better interpretability as follows.

$$
\mathbb{E}d(y,\eta)
$$
$$
= \int d(y,\eta) \frac{(\sqrt{2\pi}\sigma)^{-d}}{J_{p_0}(\mathrm{Log}_{p_0} y)} \exp\left(-\frac{\|\mathrm{Log}_{p_0} y - \mathrm{Log}_{p_0} \eta\|^2}{2\sigma^2}\right) \mathrm{d}\nu(y)
$$
$$
\leq \int \frac{\|u + \mathrm{Log}_{p_0} \eta\|}{(\sqrt{2\pi}\sigma)^d} \exp\left(-\frac{\|u\|^2}{2\sigma}\right) \mathrm{d}\lambda(u) + d(p_0,\eta),\ u = \mathrm{Log}_{p_0} y - \mathrm{Log}_{p_0} \eta
$$
$$
\leq \int \frac{\|u\|}{(\sqrt{2\pi}\sigma)^d} \exp\left(-\frac{\|u\|^2}{2\sigma}\right) \mathrm{d}\lambda(u) + 2d(p_0,\eta),\ \text{triangle inequality}
$$
$$
= \sigma\sqrt{2} \frac{\Gamma((d+1)/2)}{\Gamma(d/2)} + 2d(p_0,\eta),\ \text{since } \frac{\|u\|}{\sigma} \sim \chi_d.
$$

$\square$

Bounds (2) and (3) follows from Lemma 1 directly. Now, we prove the bounds in (4) and (5).

*Proof.* Under the assumption that $|\mathrm{Sec}_{\mathcal{M}}| < K$ for some $K \geq 0$, then by Rauch comparison theorem (Fefferman et al., 2020, Page 1082), we have

$$
\frac{\sin(\sqrt{K}r)}{\sqrt{K}r}\|\log_{p_0}(x) - \log_{p_0}(x')\| \leq d(x,x') \leq \frac{\sinh(\sqrt{K}r)}{\sqrt{K}r}\|\log_{p_0}(x) - \log_{p_0}(x')\|,
$$

for any $x, x' \in B_r(p_0)$.

It follows that

$$
\frac{\sin(\sqrt{K}r)}{\sqrt{K}r}\|\log_{p_0}(y) - \log_{p_0}(\eta)\| \leq d(y,\eta) \leq \frac{\sinh(\sqrt{K}r)}{\sqrt{K}r}\|\log_{p_0}(y) - \log_{p_0}(\eta)\|,
$$

for any $\eta, y \in B_r(p_0)$.

Under Assumption 1, we have $\eta \in B_r(p_0)$. For $y \in B_r(p_0)$, we want project $y$ back into $B_r(p_0)$. Note that this is no privacy leakage during this step, as the projection only depends on $p_0$ and $r$, which requires no privacy protection. Denote $y^*$ as the projection of $y$ back into $B_r(p_0)$, defined as

$$
y^* = \exp_{p_0}\left(\frac{r}{\|\log_{p_0} y\|}\log_{p_0} y\right).
$$

Immediately, we have $\|\log_{p_0} y^* - \log_{p_0} \eta\| \leq \|\log_{p_0} y - \log_{p_0} \eta\|$. After the projection, we have,

$$
\mathbb{E}d(\eta,y) = \int_{B_r(p_0)} d(\eta,y)\,\mathrm{d}\mathbb{P}(y) + \int_{\mathcal{M}\setminus B_r(p_0)} d(\eta,y^*)\,\mathrm{d}\mathbb{P}(y)
$$
$$
\leq \int_{B_r(p_0)} \frac{\sinh(\sqrt{K}r)}{\sqrt{K}r}\|\log_{p_0}(\eta) - \log_{p_0}(y)\|\,\mathrm{d}\mathbb{P}(y)
$$
$$
+ \int_{\mathcal{M}\setminus B_r(p_0)} \frac{\sinh(\sqrt{K}r)}{\sqrt{K}r}\|\log_{p_0}(\eta) - \log_{p_0}(y^*)\|\,\mathrm{d}\mathbb{P}(y)
$$
$$
\leq \int_{B_r(p_0)} \frac{\sinh(\sqrt{K}r)}{\sqrt{K}r}\|\log_{p_0}(\eta) - \log_{p_0}(y)\|\,\mathrm{d}\mathbb{P}(y)
$$
$$
+ \int_{\mathcal{M}\setminus B_r(p_0)} \frac{\sinh(\sqrt{K}r)}{\sqrt{K}r}\|\log_{p_0}(\eta) - \log_{p_0}(y)\|\,\mathrm{d}\mathbb{P}(y)
$$
$$
= \int_{\mathcal{M}} \frac{\sinh(\sqrt{K}r)}{\sqrt{K}r}\|\log_{p_0}(\eta) - \log_{p_0}(y)\|\,\mathrm{d}\mathbb{P}(y)
$$
$$
= \frac{\sinh(\sqrt{K}r)}{\sqrt{K}r}\mathbb{E}\|\log_{p_0}(\eta) - \log_{p_0}(y)\|.
$$

Note that from the proof for Lemma 1, we have

$$\mathbb{E}\| \log_{p_0}(y) - \log_{p_0}(\eta)\| = \sigma d, \qquad \text{for Laplace mechanism,}$$

$$\mathbb{E}\| \log_{p_0}(y) - \log_{p_0}(\eta)\| = \sigma\sqrt{2}\frac{\Gamma((d+1)/2)}{\Gamma(d/2)}, \qquad \text{for Gaussian mechanism.}$$

The result follows. $\qquad\qquad\square$

## B  BACKGROUND MATERIALS

### B.1  RIEMANNIAN GEOMETRY

Let $\mathcal{M}$ be a $d$-dimensional Riemannian manifold endowed with a Riemannian metric $g$, which assigns to each point $p \in \mathcal{M}$ a smoothly varying inner product $\langle \cdot, \cdot \rangle_p$ on the tangent space $T_p\mathcal{M}$. This inner product induces a norm $\|v\|_p = \langle v, v \rangle_p^{1/2}$, enabling the measurement of geometric quantities such as angles, lengths, and distances. For a smooth curve $\gamma(t)$ on $\mathcal{M}$, the length is given by

$$L(\gamma) = \int \|\dot{\gamma}(t)\|_{\gamma(t)}\, dt = \int \sqrt{\langle \dot{\gamma}(t), \dot{\gamma}(t) \rangle_{\gamma(t)}}\, dt.$$

Curves that locally minimize length are called geodesics, and they play a central role in defining intrinsic geometry. A Riemannian manifold is said to be geodesically complete if every geodesic can be extended to the entire real line $\mathbb{R}$; we assume this property holds throughout. Based on the definition of length, the distance between any two points $p, q \in \mathcal{M}$ is defined as the infimum of the lengths over all piecewise smooth curves joining them:

$$d(p, q) = \inf_{\gamma(0)=p, \gamma(1)=q} L(\gamma).$$

In local coordinates, the metric $g$ is represented by a positive definite matrix $g = (g_{ij})$, and the Lebesgue measure is denoted by $\lambda$. The metric tensor induces a natural volume measure $\nu$ on the Borel $\sigma$-algebra of $\mathcal{M}$, given in coordinates by $d\nu = \sqrt{|\det g|}\, d\lambda$. This Riemannian volume measure will serve as the default reference measure for integration and probability throughout the paper.

Geodesic completeness ensures that the Riemannian exponential map is globally defined. Given a point $p \in \mathcal{M}$ and a tangent vector $v \in T_p\mathcal{M}$, the geodesic $\gamma_{(p,v)}(t)$ satisfying $\gamma_{(p,v)}(0) = p$ and $\dot{\gamma}_{(p,v)}(0) = v$ exists for all $t \in \mathbb{R}$, and defines the Riemannian exponential map via $\mathrm{Exp}_p(v) = \gamma_{(p,v)}(1)$. Around each point $p$, there exists a neighbourhood $V \subset T_p\mathcal{M}$ and $U \subset \mathcal{M}$ such that the restriction $\mathrm{Exp}_p|_V : V \to U$ is a diffeomorphism. Its inverse, the Riemannian logarithmic map, is denoted by $\mathrm{Log}_p : U \to T_p\mathcal{M}$ and satisfies $\mathrm{Log}_p(q) = v$ whenever $q = \mathrm{Exp}_p(v)$. In such normal neighbourhoods, the Riemannian distance can be expressed in closed form as $d(p, q) = \|\mathrm{Log}_p(q)\|_p$, reducing the computation of distances to norms in the tangent space.

The primary focus of this paper is on Hadamard manifolds, which are simply connected complete Riemannian manifolds of non-positive curvature. It is named after the famous Cartan-Hadamard theorem which states that for any $d$-dimensional Hadamard manifold $\mathcal{M}$, it is differomorphic to $\mathbb{R}^d$ and more precisely, at any point $p \in \mathcal{M}$, the exponential mapping $\mathrm{Exp}_p : T_p\mathcal{M} \to \mathcal{M}$ is a diffeomorphism and thus $\mathrm{Log}_p$ is defined everywhere on $\mathcal{M}$. This property enables us to develop the Exponential-Wrapped mechanisms in Sections 2.2 and 2.3. Another important property of the Hadamard manifold is that $\mathrm{Log}_p$ is a contraction for any $p \in \mathcal{M}$. That is, $\|\mathrm{Log}_p q_1 - \mathrm{Log}_p q_2\|_p \le d(q_1, q_2)$ for any $p, q_1, q_2 \in \mathcal{M}$. For more technical details on Hadamard manifolds, please refer to Petersen (2006); Shiga (1984).

### B.2  DIFFERENTIAL PRIVACY

Differential privacy (DP) is a principled framework for quantifying privacy guarantees in data analysis.

**Definition B.1** ((Dwork et al., 2006a))**.** *A data-releasing mechanism $M$ is said to be $(\varepsilon, \delta)$-**DP** with $\varepsilon \ge 0, 0 \le \delta \le 1$, if for any adjacent datasets, denoted as $\mathcal{D} \simeq \mathcal{D}'$, differing in only one record, we have $\Pr(M(\mathcal{D}) \in A) \le e^\varepsilon \Pr(M(\mathcal{D}') \in A) + \delta$ for any measurable set $A$ in the range of $M$. For $\delta = 0$, $M$ is said to be $\varepsilon$-**DP**.*

Since $(\varepsilon, \delta)$-DP is a well-defined concept on any measurable space (Wasserman and Zhou, 2010), it can be readily extended to any Riemannian manifold equipped with the Borel $\sigma$-algebra. One relaxation of $\varepsilon$-DP is the Rényi DP, which is based on Rényi divergence. It shares many important properties with $\varepsilon$-DP while allowing tighter analysis of composite heterogeneous mechanisms.

**Definition B.2** ((Mironov, 2017)). *A mechanism $M$ is said to have $\varepsilon$-**Rényi Differential Privacy (RDP)** of order $\alpha$, or $(\alpha, \varepsilon)$-RDP for short, if $D_\alpha\left(M(\mathcal{D})\|M\left(\mathcal{D}'\right)\right) \leq \varepsilon$ for all neighbouring datasets $D \simeq D'$, where the Rényi divergence of a finite order $\alpha \neq 1$ is defined as*

$$D_\alpha(P\|Q) = \frac{1}{\alpha - 1} \log \mathbb{E}_{x \sim Q} \left( \frac{P(x)}{Q(x)} \right)^\alpha,$$

*and Renyi divergence at orders $\alpha = 1, \infty$ are defined by continuity.*

Another way of extending the differential privacy definition is through the viewpoint of the statistical hypothesis testing (Wasserman and Zhou, 2010; Kairouz et al., 2017). In the context of hypothesis testing, we define $H_0$: the underlying dataset is $\mathcal{D}$ and $H_1$: the underlying dataset is $\mathcal{D}'$. As the values of $\varepsilon$ and $\delta$ decrease, the task of conducting this hypothesis testing becomes more difficult. This means that detecting the presence of an individual based on the outcome of the mechanism becomes increasingly challenging. With this interpretation in mind, we can extend $(\varepsilon, \delta)$-differential privacy to Gaussian differential privacy (GDP).

Denote the outcome distribution under $H_0$ and $H_1$ as $M(\mathcal{D})$ and $M(\mathcal{D}')$, respectively. We introduce the optimal trade-off function between type I and type II errors as follows,

$$T\left(M(\mathcal{D}), M\left(\mathcal{D}'\right)\right) : [0, 1] \to [0, 1], \quad \alpha \mapsto T\left(M(\mathcal{D}), M\left(\mathcal{D}'\right)\right)(\alpha),$$

where $T\left(M(\mathcal{D}), M\left(\mathcal{D}'\right)\right)(\alpha)$ is the smallest type II error when type I error equals $\alpha$. GDP centres around this optimal trade-off function and is defined as follows.

**Definition B.3** ((Dong et al., 2022)). *A mechanism $M$ is said to satisfy $\mu$-**Gaussian Differential Privacy** ($\mu$-**GDP**) if $T\left(M(\mathcal{D}), M\left(\mathcal{D}'\right)\right) \geq G_\mu$ for all neighbouring datasets $\mathcal{D} \simeq \mathcal{D}'$ with $G_\mu := T(N(0,1), N(\mu, 1))$.*

However, the involvement of the optimal trade-off function $T(M(\mathcal{D}), M(\mathcal{D}'))$ makes Definition B.3 difficult to work with on Riemannian manifolds. To make this definition more tractable, we adapt the equivalent characterization from Jiang et al. (2023), which is based on Dong et al. (2022, Corollary 1).

**Definition B.4** (Gaussian Differential Privacy (Dong et al., 2022; Jiang et al., 2023)). *A $\mathcal{M}$-valued data-releasing mechanism $M$ is said to be $\mu$-**GDP** if it's $(\varepsilon, \delta_\mu(\varepsilon))$-DP for all $\varepsilon \geq 0$, where*

$$\delta_\mu(\varepsilon) := \Phi\left(-\frac{\varepsilon}{\mu} + \frac{\mu}{2}\right) - e^\varepsilon \Phi\left(-\frac{\varepsilon}{\mu} - \frac{\mu}{2}\right),$$

*with $\Phi$ denotes the cumulative distribution function of the standard normal distribution.*

### B.3 SPDM SPACE

Let $S_m^+$ denote the manifold of $m \times m$ real symmetric positive-definite matrices, with tangent space at each point identified with $S_m$, the space of $m \times m$ symmetric matrices. The affine-invariant (Rao-Fisher) Riemannian metric endows $S_m^+$ with non-positive sectional curvature and desirable invariance properties, but introduces substantial analytical and computational complexity. In contrast, the Log-Euclidean and Log-Cholesky metrics induce flat Riemannian geometries on $S_m^+$, each derived from a bi-invariant Lie group structure: the former using the matrix logarithm, and the latter the Cholesky decomposition. The Log-Euclidean metric defines distances via the Frobenius norm by applying the matrix logarithm Log, $\|\mathrm{Log}(p) - \mathrm{Log}(q)\|_F$, allowing closed-form expressions for geodesics and Fréchet means. The Log-Cholesky metric offers similarly explicit formulas while providing improved numerical stability and computational efficiency.

Consider the data $X_1, \ldots, X_n \in S_m^+$. Under the Log-Euclidean metric, the sample Fréchet mean has the following closed-form expression,

$$\bar{x} = \mathrm{Exp}\left\{ \frac{1}{n} \sum_{i=1}^{n} \mathrm{Log}(X_i) \right\}, \tag{8}$$

where Exp and Log denote the matrix exponential and logarithm maps.

Under the Log-Cholesky metric, the sample Fréchet mean has the following expression,

$$\bar{x} = \bar{x}^*(\bar{x}^*)^\top, \tag{9}$$

where

$$\bar{x}^* = \frac{1}{n}\sum_{i=1}^n L_i + \mathrm{Exp}\left\{\frac{1}{n}\sum_{i=1}^n \mathrm{Log}\mathbb{D}(L_i)\right\}, \tag{10}$$

with $L_i$ being the cholesky decomposition of $X_i$ such that $L_i L_i^\top = X_i$, $\lfloor \cdot \rfloor$ returning the strictly lower triangular matrix, and $\mathbb{D}(\cdot)$ returning the diagonal matrix. Refer to Arsigny et al. (2007); Lin (2019) for more details.

Although these metrics forgo affine invariance, their flatness simplifies analysis and makes them particularly suitable for statistical inference and privacy-preserving tasks. In the simulations that follow, we compare all three metrics, with particular emphasis on the Log-Euclidean and Log-Cholesky approaches due to their practical advantages. For more details on SPDM spaces and these metrics, refer to Arsigny et al. (2007); Lin (2019); Said et al. (2017); Reimherr et al. (2021).

### B.4 HYPERBOLIC SPACE

Hyperbolic space is a space of constant negative curvature. Here, we will focus on the Lorentz model $\mathbb{H}_d$, also referred to as the hyperboloid model, of hyperbolic space. For the Lorentz model $\mathbb{H}_d$, each point is identified with $x \in \mathbb{R}^{d+1}$ such that $\langle x, x \rangle_L = -1$ with the Lorentz inner product defined as follows,

$$\langle x, y \rangle_L = x_0 y_0 + \sum_{i=1}^n x_i y_i.$$

The distance between two points $x, y \in \mathbb{H}_d$ is then defined as,

$$d_L(x, y) = \mathrm{arccosh}(-\langle x, y \rangle_L).$$

The tangent space $T_x\mathbb{H}_d$ at each point $x \in \mathbb{H}_d$ is identified as $\{u : \langle u, x \rangle_L = 0\}$. The exponential map $\exp_x$ has the following closed-form expression,

$$\exp_x(u) = \cosh(\|u\|_L)x + \sinh(\|u\|_L)\frac{u}{\|u\|_L},$$

with $\|u\|_L = \sqrt{\langle u, u \rangle_L}$. Similarly, the logarithm map has the following expression,

$$\log_x(y) = \frac{\mathrm{arccosh}(\alpha)}{\sqrt{\alpha^2 - 1}}(y - \alpha x),$$

with $\alpha = -\langle x, y \rangle_L$. For more details on the Lorentz model, see Nagano et al. (2019); Cho et al. (2022).

## C SAMPLING FROM EXPONENTIAL-WRAPPED DISTRIBUTION

### C.1 EXPONENTIAL-WRAPPED LAPLACE DISTRIBUTION

The EWL Distribution is the push-forward probability of the tangent space probability defined by the probability density $h(u) \propto \exp\{-\|u - \log_{p_0} \eta\|/\sigma\}$. The sampling procedure for the EWL Distribution is straightforward:

1. Sampling from $u \sim h(u) \propto \exp\{-\|u - \log_{p_0} \eta\|/\sigma\}$.
2. Computing $\exp_{p_0} u$.

Note that the sampling step $u \sim h(u)$ needs some clarification. We want to emphasize that $\|\cdot\|$ within step 1 is not the $l_2$-norm but rather the norm induced by the Riemannian metric $g_p$. Note that the tangent space $T_p\mathcal{M}$ equipped with $g_p$ can be identified with $\mathbb{R}^d$ equipped with the Euclidean metric

as follows. Let $\{e_1, \ldots, e_d\}$ be an orthogonal basis w.r.t. $g_p$ on $T_p\mathcal{M}$, then any point $v \in T_p\mathcal{M}$ can be identified with a point $v^* \in \mathbb{R}^d$ via the following map,

$$\iota_p : T_p\mathcal{M} \to \mathbb{R}^d, \quad v = \sum_{i=1}^{d} a_i e_i \mapsto \iota_p(v) = (a_1, \ldots, a_d), \tag{11}$$

Denote this map as $\iota_p : T_p\mathcal{M} \to \mathbb{R}^d$, and note that $\iota_p$ is a isometry for any $p \in \mathcal{M}$ as,

$$g_p(v_1, v_2) = \iota_p(v_1)^\top \iota_p(v_2) = \langle \iota_p(v_1), \iota_p(v_2) \rangle,$$

where $\langle \cdot, \cdot \rangle$ denote the euclidean inner product here. Putting it together, we have the following sampling procedure,

1. Sampling from $u \sim h(u) \propto \exp\{-\|u - \iota_{p_0}(\log_{p_0} \eta)\|_2/\sigma\}$.
2. Computing $\exp_{p_0}(\iota_{p_0}^{-1}(u))$.

## C.2 Exponential-Wrapped Gaussian Distribution

Implementing the EWG mechanism for $(\varepsilon, \delta)$-DP is straightforward. We follow a similar procedure as in Algorithm 1. After determining the appropriate $\sigma$ numerically from inequality (1)—using a method such as that proposed in Balle and Wang (2018b)—one can proceed by

1. first sampling $\mathbf{u}$ from the multivariate Gaussian distribution $\mathcal{N}(\mathbf{0}, \sigma^2 \mathbf{I}_d)$.
2. The privatized summary is then computed as

$$\exp_{p_0}\left\{\iota_{p_0}^{-1}(\mathbf{u}) + \iota_{p_0}[\log_{p_0}(f(\mathcal{D}))]\right\}.$$

Suppose $\mathcal{M}$ is the space of SPDM equipped with log-Euclidean metric, the EWG mechanism with footpoint $p_0 = \mathbf{I}$ reduces to the tangent Gaussian mechanism in Utpala et al. (2023b). Hence, the EWG mechanism is a generalization of the tangent Gaussian mechanism, as our mechanism can be employed for any Hadamard manifold equipped with any Riemannian metric. This makes our EWG mechanism the first working mechanism to achieve $(\varepsilon, \delta)$-DP in SPDM under the non-log-Euclidean metric.

The implementation of the EWG mechanism for $\mu$-GDP is similar.

## C.3 Sampling From EWG on SPDM Space

Here, we discuss how to sample from the EWG distribution on SPDM space equipped with the three different metrics. All the sampling procedures are summarized in Algorithm 2.

**(i) Affine-Invariant metric** We note that the Riemannian metric $g_P$ on $T_P S_m^+$ is defined as,

$$g_P(X, Y) = \mathrm{trace}(P^{-1}XP^{-1}Y).$$

Due to the affine invariant property of the Affine-Invariant metric, we have,

$$\begin{aligned} g_P(X, Y) &= g_{\mathbf{I}_m}(P^{-1/2}XP^{-1/2}, P^{-1/2}YP^{-1/2}) \\ &= \mathrm{trace}(P^{-1/2}XP^{-1/2}P^{-1/2}YP^{-1/2}) \\ &= \langle P^{-1/2}XP^{-1/2}, P^{-1/2}YP^{-1/2} \rangle_F, \end{aligned}$$

where $\langle \cdot, \cdot \rangle_F$ denotes the Frobenius inner product. Note $P^{-1/2}XP^{-1/2} \in S_m$, the space of $m \times m$ symmetric matrices, we can map them into $\mathbb{R}^{m(m+1)/2}$ via the function $\mathrm{vecd} : S_m \to \mathbb{R}^{m(m+1)/2}$ which is defined as

$$\mathrm{vecd}(W) = (\mathrm{diag}(W)^\top, \sqrt{2}\,\mathrm{offdiag}(W)^\top)^\top,$$

where $\mathrm{diag}(W)$ / is an $m$-dimensional vector containing the diagonal entries of $W$ and $\mathrm{offdiag}(Y)$ is an $m(m-1)/2$-dimensional vector containing the off-diagonal entries of $W$ copied from below the diagonal columnwise (or above the diagonal row-wise). The inclusion of the factor $\sqrt{2}$ for the off diagonal entries ensure that ,

$$\langle X, Y \rangle_F = \mathrm{vecd}(X)^\top \mathrm{vecd}(Y),$$

for any $X, Y \in S_m$. See Schwartzman (2006; 2016) for more details on this vectorization operator. It follows that the map $\iota_P$ defined as

$$\iota_P : T_p S_m^+ \to \mathbb{R}^{m(m+1)/2}, \quad X \mapsto \text{vecd}(P^{-1/2} X P^{-1/2}) \tag{12}$$

is an isometry. Thus, to sample from EWG with footpoint $p_0$, center $\eta$, and rate $\sigma > 0$ under the Affine-Invariant metric can be summarized as follows.

1. Map the center to $\mathbb{R}^{m(m+1)/2}$ via $\log_{p_0}$ and $\iota_{p_0}$ as $\iota_{p_0}[\log_{p_0}(\eta)]$.

2. Sample $\mathbf{u} \sim \mathcal{N}(\iota_{p_0}[\log_{p_0}(\eta)], \sigma^2 \mathbf{I}_d)$.

3. Map $\mathbf{u}$ back to $\mathcal{M}$ via $\iota_{p_0}^{-1}$ and $\exp_{p_0}$ as $\exp_{p_0}\{\iota_{p_0}^{-1}(\mathbf{u})\}$.

**(ii) Log-Euclidean metric** For both Log-Euclidean and Log-Cholesky metric, we fix the footpoint to be $\mathbf{I}_m$ as the footpoint will have no impact on the result due to vanishing curvature and $\mathbf{I}_m$ simplifies the computation a bit. We note that under the Log-Euclidean metric, we have,

$$g_{\mathbf{I_m}}(X, Y) = \text{trace}(XY) = \langle X, Y \rangle_F.$$

Thus, follows a similar argument as in the Affine-invariant case and note that vecd is a isometry between $T_{\mathbf{I}_m} S_m^+$ and $\mathbb{R}^{m(m+1)/2}$, we can sample from EWG with footpoint $p_0$, center $\eta$, and rate $\sigma > 0$ under the Log-Eucldiean metric can be summarized as follow.

1. Map the center $\eta$ to $\mathbb{R}^{m(m+1)/2}$ via Log and vecd as $\text{vecd}[\text{Log}(\eta)]$.

2. Sample $\mathbf{u} \sim \mathcal{N}(\text{vecd}[\text{Log}(\eta), \sigma^2 \mathbf{I}_d)$.

3. Map $\mathbf{u}$ back to $\mathcal{M}$ via $\text{vecd}^{-1}$ and Exp as $\text{Exp}\{\text{vecd}^{-1}(\mathbf{u})\}$.

Note that Exp and Log denote the matrix exponential and logarithm, respectively.

**(iii) Log-Euclidean metric** We note the following relation,

$$S_m^+ \xrightarrow{\mathscr{L}} \mathcal{L}^+ \xrightarrow{\widetilde{\text{Log}}_{\mathbf{I}_m}} \mathcal{L} \xrightarrow{\widetilde{\text{vecd}}} \mathbb{R}^{m(m+1)/2},$$

where

1. $\mathcal{L}$ denotes the space of upper triangular matrices,

2. $\mathcal{L}^+$ denotes the space of upper triangular matrices with postive diagonal entries,

3. $\mathscr{L}$ denotes the Log-Cholesky decomposition,

4. $\widetilde{\log}$ is defined as,

$$\widetilde{\log}_L(K) = \lfloor K \rfloor - \lfloor L \rfloor + \mathbb{D}(L)\text{Log}\{\mathbb{D}(L)^{-1}\mathbb{D}(K)\},$$

5. and the operator $\widetilde{\text{vecd}}$ is defined as follow,

$$\widetilde{\text{vecd}}(X) = (\text{diag}(X)^\top, \text{offdiag}(x)^\top)^\top \quad \text{for } X \in \mathcal{L}.$$

Once again, we have $\langle X, Y \rangle_F = \widetilde{\text{vecd}}(X)^\top \widetilde{\text{vecd}}(Y)$ for any $X, Y \in \mathcal{L}$, and thus $\widetilde{\text{vecd}}$ is a isometry between $\mathcal{L}$ and $\mathbb{R}^{m(m+1)/2}$. Combine with fact that $\widetilde{\log}_{\mathbf{I}_m} \circ \mathscr{L}$ is a isometry between $S_m^+$ and $\mathcal{L}$, we have

$$\widetilde{\text{vecd}} \circ \widetilde{\log}_{\mathbf{I}_m} \circ \mathscr{L}$$

is a isometry between $S_m^+$ and $\mathbb{R}^{m(m+1)/2}$. Thus, to sample from EWG with footpoint $p_0$, center $\eta$, and rate $\sigma > 0$ under the Log-Cholesky metric can be summarized as follow.

1. Map the center $\eta$ to $\mathbb{R}^{m(m+1)/2}$ as

$$\widetilde{\text{vecd}} \circ \widetilde{\log}_{\mathbf{I}_m} \circ \mathscr{L}(\eta).$$

2. Sample

$$\mathbf{u} \sim \mathcal{N}\left\{\widetilde{\text{vecd}} \circ \widetilde{\log}_{\mathbf{I}_m} \circ \mathscr{L}(\eta), \sigma^2 \mathbf{I}_d\right\}.$$

3. Map $\mathbf{u}$ back to $\mathcal{M}$ as

$$\left[\widetilde{\text{vecd}} \circ \widetilde{\log}_{\mathbf{I}_m} \circ \mathscr{L}\right]^{-1}(\mathbf{u}).$$

---

**Algorithm 2** Generate GDP Fréchet mean on SPDM space

---

**Input:** Data radius $r$, privacy budget $\mu$, private data $X_1, \ldots, X_n \in S_m^+$, Riemannian Metric $\rho$, footpoint $p_0$ for Affine-Invariant metric.
**Output:** Privatized Fréchet mean $\tilde{x}_{\text{EWG}}^{\text{gdp}} \in S_m^+$.

1: **Sample** $\mathbf{v} \sim \mathcal{N}(\mathbf{0}, \mathbf{I}_d)$ with $d = m(m+1)/2$.
2: **if** $\rho = \rho^{\text{LE}}$ **then**
3:     **Compute** sample Fréchet mean $\bar{x}$ using equation 8.
4:     **Compute** $\tilde{x}_{\text{EWG}}^{\text{gdp}} = \text{Exp}\left(\text{Log}(\bar{x}) + \text{vecd}^{-1}(\sigma \mathbf{v})\right)$ with $\sigma = 2r_0/\mu$.
5: **else if** $\rho = \rho^{\text{LC}}$ **then**
6:     **Compute** sample Fréchet mean $\bar{x}$ using equation 9.
7:     **Compute** $L = \widetilde{\exp}_{\mathbf{I}_m}\left(\widetilde{\log}_{\mathbf{I}_m} \circ \mathscr{L}(\bar{x}) + \widetilde{\text{vecd}}^{-1}(\sigma \mathbf{v})\right)$ with $\sigma = 2r_0/\mu$.
8:     **Compute** $\tilde{x}_{\text{EWG}}^{\text{gdp}} = LL^\top$.
9: **else if** $\rho = \rho^{\text{AI}}$ **then**
10:     **Compute** sample Fréchet mean $\bar{x}$ using Gradient Descent algorithm.
11:     **Compute** $\tilde{x}_{\text{EWG}}^{\text{gdp}} = \exp_{p_0}\left\{\iota_{p_0}^{-1}\left(\iota_{p_0}[\log_{p_0}(\bar{x})] + \sigma \mathbf{v}\right)\right\}$ where $\iota_{p_0}$ is defined in equation 12.
12: **end if**
13: **Return**: $\tilde{x}_{\text{EWG}}^{\text{gdp}}$.

---

### C.4 SAMPLING FROM EWG ON HYPERBOLIC SPACE

To sample from EWG with footpoint $p_0$, center $\eta$, and rate $\sigma > 0$ on $\mathbb{H}_d$, we modifies the approach described in Cho et al. (2022), which is stated below:

1. Map the center $\eta$ to $T_{p_0}\mathbb{H}_d$ as $\log_{p_0}(\eta)$.

2. Sample $\mathbf{u} \sim \mathcal{N}(0, \sigma^2 \mathbf{I}_d)$ and parallel transport the vector $[0, \mathbf{u}]$ to the tangent space $T_{p_0}\mathbb{H}_d$,

$$\tilde{\mathbf{u}} = \text{PT}_{e_1 \to p_0}([0, \mathbf{u}]).$$

3. Map $\tilde{\mathbf{u}} + \log_{p_0}(\eta)$ back to $\mathbb{H}_d$ via exponential map,

$$\exp_{p_0}(\tilde{\mathbf{u}} + \log_{p_0}(\eta)).$$

## D COMPUTATION TIME COMPARISON

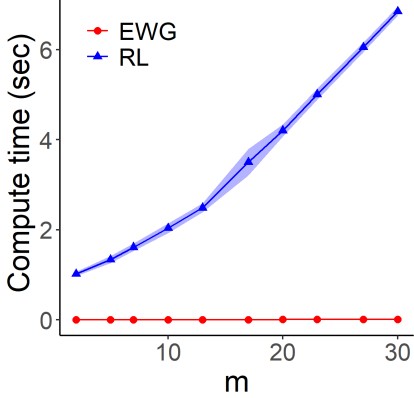

Figure 4: Computation time comparison of EWG and RL Mechanisms under the affine-invariant metric. The blue line with triangular symbols represents the RL mechanism, while the red line with circular symbols represents the EWG mechanism. The RL mechanism is implemented with a burn-in size of 10,000.

Table 1: Computation time (seconds) comparison between RL and EWG mechanisms under the affine-invariant metric. The RL mechanism is implemented with a burn-in size of 10,000. Results from 10 Monte Carlo replications.

| | mechanism | | | |
| | EWG | | RL | |
| size $m$ | mean | SD | mean | SD |
|---|---|---|---|---|
| 2 | 0.00248 | 0.00252 | 1.01774 | 0.05029 |
| 5 | 0.00160 | 0.00022 | 1.33547 | 0.07959 |
| 7 | 0.00166 | 0.00011 | 1.61042 | 0.08033 |
| 10 | 0.00204 | 0.00032 | 2.03352 | 0.10878 |
| 13 | 0.00241 | 0.00018 | 2.48609 | 0.09874 |
| 17 | 0.00329 | 0.00083 | 3.50102 | 0.29508 |
| 20 | 0.00402 | 0.00067 | 4.20360 | 0.13551 |
| 23 | 0.00460 | 0.00065 | 5.00708 | 0.12180 |
| 27 | 0.00761 | 0.00509 | 6.05043 | 0.10942 |
| 30 | 0.00709 | 0.00074 | 6.85141 | 0.10098 |

# E  SIMULATION WITH VARYING SAMPLE SIZES

Supplemental to the results we have in Section 4, we examine the effect of sample size $n$ on performance. Here, we repeat the same simulation for Hyperbolic space but fix the privacy budget $\mu$ at 0.1 while varying the sample size $n \in 10, 20, \ldots, 100$. The results are provided in Figure 5.

As the sample size $n$ only factors into the result via the computation of the sensitivity. The sensitivity for the sample Fréchet mean under Hadamard manifolds takes the form of $\Delta = 2r/n$ and the rate parameter $\sigma$ of the noises injected takes the form of $\sigma = \Delta/\mu = 2r/(n\mu)$. Thus, one would expect $n$ and $\mu$ to have the same effect on the result. Indeed, this is what is observed in Figure 5, which mirrors what we observed in Figure 2.

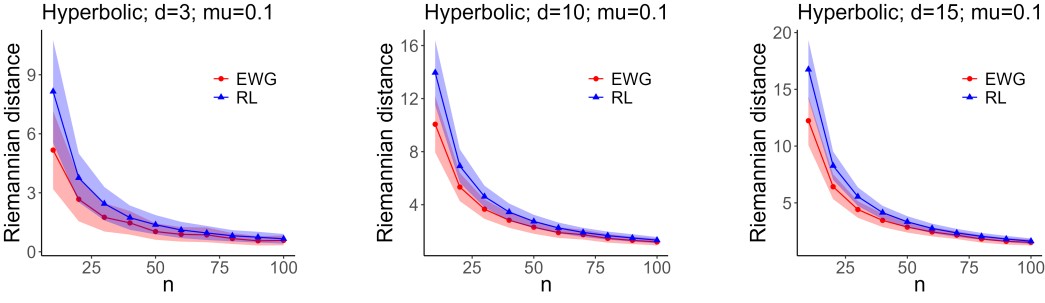

Figure 5: Utility Comparison of EWG and RL Mechanisms on Hyperbolic space $\mathbb{H}_d$. Blue lines with triangular symbols show the Riemannian distances $d(\bar{x}, \tilde{x}_{\mathrm{RL}}^{\mathrm{gdp}})$, and the red line with circular symbols represent the Riemannian distances $d(\bar{x}, \tilde{x}_{\mathrm{EWG}}^{\mathrm{gdp}})$.

# F  EXPERIMENT WITH VARYING FOOTPOINTS

Supplemental to the results we have in Section 4, we examine the effect of footpoint on performance. Here, we repeat the same simulation for Hyperbolic space but we randomly select a point within $B_{r/2}(p_0)$ as the footpoint for each simulation. The results are provided in Figure 6.

Compare to the result in Figure 2, we observed the performance of EWG mechanism under randomly selected footpoint is slightly worse but still outperform the RL mechanism.

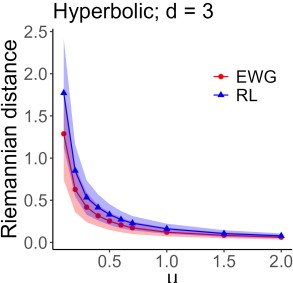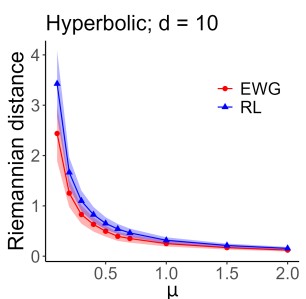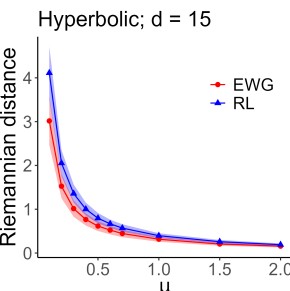

Figure 6: Utility Comparison of EWG and RL Mechanisms on Hyperbolic space $\mathbb{H}_d$. Blue lines with triangular symbols show the Riemannian distances $d(\bar{x}, \tilde{x}_{\text{RL}}^{\text{gdp}})$, and the red line with circular symbols represent the Riemannian distances $d(\bar{x}, \tilde{x}_{\text{EWG}}^{\text{gdp}})$.

## G   SELECTING A DATA-DEPENDENT FOOTPOINT

Mentioned in 3.2, if there is no prior knowledge, it might be worthwhile to spend part of the privacy budget to select a data-dependent footpoint $p_0$. In the context of outputting sample fréchet mean, a natural candidate to use would be the extrinsic sample fréchet mean $\bar{x}_E$, see Bhattacharya and Patrangenaru (2003, Section 3) for details. The extrinsic frechet mean $\bar{x}_E$ is obtained by

1. Embedding the manifold $\mathcal{M}$ into a ambient Euclidean space via the embedding $j : \mathcal{M} \to \mathbb{R}^k$: $j(X_1), \ldots, j(X_n)$.

2. Compute the Euclidean mean of $j(X_1), \ldots, j(X_n)$ as $\overline{j(x)}$.

3. Project the mean $\overline{j(x)}$ back to $j(\mathcal{M})$ via the projection $P$: $P(\overline{j(x)})$.

4. Lastly, map back to $\mathcal{M}$ by reverse the embedding: $\bar{x}_E = j^{-1}(P(\overline{j(x)}))$.

To obtain a differentially private version of $\bar{x}_E$, we can simply inject the Euclidean Gaussian noise into $\overline{j(x)}$ to obtain $\overline{j(x)}^{\text{gdp}}$ via the Gaussian mechanism for GDP on Euclidean space. It follows that $\bar{x}_E^{\text{dp}} = j^{-1}(P(\overline{j(x)}^{\text{dp}}))$ is differential private by the post-processing property.

Note that this approach could extend to other $\mathcal{M}$-statistics as long as there exists a Euclidean counterpart, which is often the case in manifolds.

## H   COMPUTING RESOURCES

For simulations in section 4.1, refer to `simulation_gaussian.R` and `spd_functions.R` for the affine invariant metric, `simulation_gaussian_le.R` and `spd_functions_le.R` for Log-Euclidean metric, and `simulation_gaussian_lc.R` and `spd_functions_lc.R` for Log-Euclidean metric. Similarly, `simulation_gaussian_hyperbolic.R` and `hyperbolic_functions.R` are used for generating the results for hyperbolic space.

`GDP_plot.R`, `GDP_le_plot.R` and `GDP_lc_plot.R` are for generating the result plots in Figure 1, while as GDP_hyperbolic_plot.R are for generating the result plots in Figure 2.

For the computation time comparison, refer to `simulation_gaussian_time.R` and `GDP_time_plot.R`.

For the experiments on `OCTMNIST` dataset in Section 4.2, refer to `octmnist_data.R` for generating covariance descriptors, `octmnist_gaussian.R` for simulation on the covariance descriptors, and `octmnist_GDP_plot.R` for generating the result plots in Figure 3.

The simulations were performed using R on a PC with a 12th Gen Intel Core i5-12600K CPU with 32 GB of RAM running Windows 11. Computation times for EWG and RL mechanisms are given in Table 1 and Figure 4.

