# OpenReview forum: "Exponential-Wrapped Mechanisms: Differential Privacy on Hadamard Manifolds Made Practical"
_ICLR.cc/2026/Conference — ICLR 2026 Poster_

### Official Review · Reviewer_G5je · 2025-10-30

**Soundness:** 3
**Presentation:** 3
**Contribution:** 3
**Rating:** 8
**Confidence:** 3

**Summary:**

The paper introduces "Exponential-Wrapped" mechanisms for achieving differential privacy on curved geometric spaces (Hadamard manifolds) like hyperbolic space and symmetric positive definite matrices used in medical imaging. Instead of using slow MCMC sampling like existing methods, they simply sample from distributions in flat tangent space and map them to the manifold using the exponential map. This approach is computationally efficient, works for multiple types of differential privacy (ε-DP, GDP, RDP), and achieves better utility-privacy tradeoffs than previous Riemannian privacy methods. They demonstrate strong performance on both synthetic and real-world data, particularly for high-dimensional medical imaging applications.

**Strengths:**

1) The paper tackles differential privacy for manifold-valued medical data (diffusion tensor imaging, OCT scans) where traditional Euclidean methods fail due to geometric incompatibility. This is increasingly important as healthcare AI systems require both geometric fidelity for accuracy and rigorous privacy guarantees for patient data protection.

2) The EWG mechanism achieves runtime improvements of several orders of magnitude over existing MCMC-based methods, with the speedup increasing in higher dimensions. This makes differential privacy practically feasible for real-world medical imaging applications where previous methods were computationally prohibitive, especially in high-dimensional SPDM spaces.

3) Comprehensive experiments span synthetic data on three different SPDM metrics and hyperbolic space, plus real-world OCTMNIST medical imaging data. The EWG mechanism consistently demonstrates superior utility across 100 Monte Carlo replications, multiple dimensions (d ∈ {3, 10, 15}), and wide privacy budget ranges, with particularly strong performance in high-dimensional regimes where it matters most.

**Weaknesses:**

1) Performance degrades on affine-invariant metric at low dimensions (d=3) with high privacy budgets due to footpoint misalignment; paper acknowledges this but provides no principled solution for choosing footpoint $p_{0}$

2) Only tests mean estimation; no evaluation on other statistical tasks like principal geodesic analysis or regression despite these being mentioned as important applications

**Questions:**

NA

---

> ### Author Response · Authors · 2025-11-19
>
> Dear reviewer G5je: Thank you very much for your valuable and constructive feedback and your great support. Below, we address each of the concerns point-to-point.
>
> > Performance degrades on affine-invariant metric at low dimensions ($d=3$) with high privacy budgets due to footpoint misalignment; paper acknowledges this but provides no principled solution for choosing footpoint $p_0$
>
> Regarding the reviewer’s concern about the differentially private procedure for selecting $p_0$. In the revised manuscript, we provide an explicit DP mechanism for estimating the footpoint. This procedure is summarized at the top of the Revision Outline and formally detailed in Appendix G of the updated PDF. We hope this resolves the concern and clarifies how $p_0$ can be selected in a fully differentially private manner.
>
> > Only tests mean estimation; no evaluation on other statistical tasks like principal geodesic analysis or regression despite these being mentioned as important applications.
>
> Thank you for pointing this out. While our current work focuses on differentially private Fréchet mean estimation as a foundational problem, we agree that extending the methodology to tasks such as principal geodesic analysis or regression is a valuable direction.

---

### Official Review · Reviewer_Lgao · 2025-10-30

**Soundness:** 3
**Presentation:** 3
**Contribution:** 3
**Rating:** 8
**Confidence:** 4

**Summary:**

The authors present a framework for achieving differential privacy with a summary taking values over Hadamard Manifolds (complete, connected Riemannian manifold with non-positive sectional curvature).  The authors achieve this by introducing the "exponential wrapped mechanism", which sanitizes the statistic on a tangent plane before mapping back to the manifold.  They demonstrate that this mechanism can be used to achieve basically any popular notion of DP (pure, approximate, gaussian, and renyi).  The authors provide an interesting theoretical analysis, bounding the noise injected for privacy.  In the case that the sectional curvature is bounded, they show that the noise scales like (1/n) which matches the results for Euclidean geometry.

**Strengths:**

The results are timely as machine learning on manifolds and with privacy are both active fields.  The framework presented by the authors is fairly complete and the mathematical results are interesting.  Extensive numerical work is presented to highlight the strengths of the approach (both in terms of accuracy and computation).

**Weaknesses:**

The major weakness of the method is that a footpoint is required to implement the methodology and this footpoint is currently not data driven (for privacy reasons).  This is reasonable for a new method and acknowledged by the authors, but the paper suffers from not having any simulations showing how sensitive their mechanism is to the choice of footpoint (even in the appendix would suffice).

The other weakness is the limit to non-positive curvature.  Though this is fairly common given how different manifolds are with non-positive and non-negative curvature.

I don't view either of these weaknesses as fatal, though the first one would be fairly easy to address.

**Questions:**

See weaknesses.

---

> ### Author Response · Authors · 2025-11-19
>
> Dear reviewer Lgao: Thank you very much for your valuable and constructive feedback and your great support. Below, we address each of the concerns point-to-point.
>
> > The major weakness of the method is that a footpoint is required to implement the methodology and this footpoint is currently not data driven (for privacy reasons). This is reasonable for a new method and acknowledged by the authors, but the paper suffers from not having any simulations showing how sensitive their mechanism is to the choice of footpoint (even in the appendix would suffice).
>
> Thank you for the suggestion. To examine the effect of footpoint selection in our mechanism, we have added additional simulations in Appendix F in the updated PDF file, where we repeat the simulation for hyperbolic space but randomly select a point within the ball $B_{r/2}(p)$ with $p = (1, 0, \dots, 0)$ as the footpoint. The results are shown in Figure 6. Compared to Figure 2, we observe that the performance of our EWG mechanism degrades slightly but still outperforms the RL mechanism. See the updated PDF file for more details.
>
>
> > The other weakness is the limit to non-positive curvature. Though this is fairly common given how different manifolds are with non-positive and non-negative curvature.
>
>
> We agree that our current framework is limited to manifolds with non-positive sectional curvature, which ensures the existence of the  push-forward of probability measures that are supported on the whole tangent space. Extending the methodology to positively curved or general Riemannian manifolds is an important and nontrivial direction for future work.

---

### Official Review · Reviewer_bvfn · 2025-10-30

**Soundness:** 3
**Presentation:** 3
**Contribution:** 3
**Rating:** 6
**Confidence:** 4

**Summary:**

In this paper the authors focus on wrapped distributions (distributions defined on tangent spaces and pushed onto the manifold via the exponential map) for Hadamard manifolds to achieve differential privacy.  The authors propose two mechanisms and show how they achieve a variety of DP definitions over many example manifolds. The insight of this paper in Theorem 3.1 shows how designation of the footprint of the mechanism directly effects the statistical utility.

**Strengths:**

The authors are thorough on manifolds and SOTA definitions of privacy.
The authors focus on:
-two mechanisms (Laplace and Gaussian)
-SPDM manifold under three metrics (affine-invariant, log-Euclidean, Log-Cholesky)
-Hyperbolic space manifold
-three definitions of privacy ( $\epsilon,\delta$-DP, Gaussian DP, and Renyi DP).

Further the authors implement their methodology under varying dimension sizes.

**Weaknesses:**

A weakness is the lack of focus on varying sample sizes. While I understand that for a fixed sample size one can see how the dimension effects utility (Fig 1.) and hence, in a sense, verifies Theorem 3.1, some experiments on what happens as n increases would be useful.

Theorem 3.1 also needs to be reworked. See questions below.

There are some typos here and there which should be fixed. For instance, many of the citation styles are incorrect in the opening paragraph.

Sometimes $p_0$ is called the footPOINT other times footPRINT. This is one example, but other notations are inconsistent

**Questions:**

I am having a difficult time understanding the second half of Theorem 3.1. First, does $|Sec_{\mathcal{M}}|$ refer to the determinant or norm of the sectional curvature? The added confusion here is that $K\geq 0$ but this paper is about Hadamard manifolds which have non positive curvature.
(The footnote is poorly placed, at first read the 2 looked like a power and hence $K^2$. Further the footnote itself uses $m$ rather than $\mathcal{M}$)
This is further confusing in line 284 where $K$ is referred to as an upper bound. Perhaps we should have $K\leq0$?

I am not convinced Line 138 is correct "our method only requires a rate of... across all manifolds." It seems the theory is limited to Hadamard manifolds, which clearly is not ALL manifolds.

The sampling section in the appendix should have exp as the second step not log, correct?

---

> ### Author Response · Authors · 2025-11-19
>
> Dear reviewer bvfn: Thank you very much for your valuable and constructive feedback. We sincerely appreciate the opportunity to clarify and improve our work. Below, we address each of the concerns point-to-point
>
> > A weakness is the lack of focus on varying sample sizes. While I understand that for a fixed sample size one can see how the dimension effects utility (Fig.1) and hence, in a sense, verifies Theorem 3.1, some experiments on what happens as $n$ increases would be useful.
>
> Thanks for the suggestion. We have added additional simulation in Appendix E (see the updated PDF file), where we repeat the simulation for hyperbolic space but fix the privacy budget $\mu = 0.1$ while varying the sample size from $n=10$ to $n=90$. The results are provided in Figure 5. As the sample size $n$ only factors into the mechanism via the computation of the sensitivity. The sensitivity for the sample Fr\'echet mean under Hadamard manifolds takes the form of $\Delta = 2r/n$ and the rate parameter $\sigma$ of the noises injected takes the form of $\sigma = \Delta/ \mu = 2r/(n\mu)$. Thus, one would expect $n$ and $\mu$ to have the same effect on the result. Indeed, this is what is observed in Figure 5, which mirrors what we observed in Figure 2. See the updated PDF file for more details.
>
> > There are some typos here and there which should be fixed. For instance, many of the citation styles are incorrect in the opening paragraph.
>
> The miss-uses of \\citet\{\}} in the first two paragraphs have been fixed; \\citet\{\}'s have been been changed into \\citep\{\}'s.
>
> > Sometimes $p_0$ is called the footPOINT other times footPRINT. This is one example, but other notations are inconsistent.
>
> Thank you for catching that. It should be all "footpoint". The use of "footprint" in lines 134, 174, 192, 201, 249, 254, 722, 808, and 1105 has been changed to "footpoint". Other notation inconsistencies have been fixed as well. See the "Notation and Typos" section in the top comment.
>
> > I am having a difficult time understanding the second half of Theorem 3.1. First, does $| \mathrm{Sec}_{\mathcal{M}} |$ refer to the determinant or norm of the sectional curvature? The added confusion here is that $K \ge 0$, but this paper is about Hadamard manifolds, which have nonpositive curvature. (The footnote is poorly placed, at first read the 2 looked like a power and hence $K^2$. Further the footnote itself uses $m$ rather than $\mathcal{M}$.) This is further confusing in line~284 where $K$ is referred to as an upper bound. Perhaps we should have $K \le 0$?
>
> We acknowledge that the use of $|\operatorname{Sec}_{\mathcal{M}}| < K$ with $K \ge 0$ was unclear and confusing. The $| \cdot|$ refers to the absolute value. Since we are dealing with Hadamard manifolds, we agree that the usage here is redundant and confusing. For better readability, it has been changed into "$\operatorname{Sec}_{\mathcal{M}} > K$ with $K <=0$" for better readability. The use of "upper bound" in line 284 will be changed to "lower bound" instead. Furthermore, the footnote has been moved to the end of the sentence, and the typo has been fixed.
>
>
> > I am not convinced Line~138 is correct "our method only requires a rate of... across all manifolds." It seems the theory is limited to Hadamard manifolds, which clearly is not all manifolds.
>
> Yes, it should be "all Hadamard manifolds" instead of "all manifolds". Thanks for pointing it out.
>
>
> > The sampling section in the appendix should have $\operatorname{exp}$ as the second step not $\operatorname{log}$, correct?
>
> Yes, the $\operatorname{log}$'s in line 1076, 1093 and 1103 should be the $\operatorname{exp}$'s instead. Thanks for pointing it out.

---

> > ### Comment · Reviewer_bvfn · 2025-11-20
> >
> > Thank you for the thorough rebuttals and the revised version of the paper. The authors have addressed my major concerns and so I have adjusted my score accordingly.

---

> > > ### Author Response · Authors · 2025-11-20
> > >
> > > Dear reviewer bvfn: Thank you very much for your support and for taking the time to review our revision. We truly appreciate your constructive feedback and are glad that the revisions have addressed your concerns.

---

### Official Review · Reviewer_w921 · 2025-11-05

**Soundness:** 3
**Presentation:** 3
**Contribution:** 3
**Rating:** 4
**Confidence:** 2

**Summary:**

The paper introduces Exponential wrapped mechanism for DP on Hadamard manifolds. As these manifolds admit a global exponential map and Lipschitz log map, the Exponential wrapped mechanism draws noise in a Euclidean tangent space and push it forward to the manifolds via Exp. This allows efficient sampling Laplace, and Gaussian noise without MCMC.  They provide utility bounds and some simulation results.

**Strengths:**

- The observation of using exponential map on Hadamard manifolds give us a clean geometry-aware DP mechanisms, and avoid MCMC burden.
- The paper also provide results beyond pure DP (approximate DP, GDP, and RDP)

**Weaknesses:**

- The utility guarantees are weak on curved manifolds.  The bound for Frechet mean contains a error term that depends on the footprint $d(p_0, \bar{x})$ which does not go to zero with the number of data $n$ unless $p_0 = \bar{x}$.  Though the paper suggests privately estimating $p_0$, it does not provide a concrete DP procedure for selecting $p_0$.


Minor comment:
- the definition of exponential wrapped mechanism can be clearer.  Is $\eta$ in definition 1 the original output $f(D)$?
- Should be footpoint or footprint?

**Questions:**

Can you provide a concrete DP mechanism for p_0 selection?

---

> ### Author Response · Authors · 2025-11-19
>
> Dear reviewer w921: Thank you very much for your valuable and constructive feedback. We sincerely appreciate the opportunity to clarify and improve our work. Below, we address each of the concerns point-to-point:
>
> > The utility guarantees are weak on curved manifolds. The bound for Fr\'echet mean contains an error term that depends on the footprint $d(p_0, \bar{x})$, which does not go to zero with the number of data $n$ unless $p_0 = \bar{x}$. Though the paper suggests privately estimating, it does not provide a concrete DP procedure for selecting $p_0$.
>
> Regarding the error term in our utility bound. The term $d(p_0, \bar{x})$ in Theorem 3.1 arises from an application of the triangle inequality. We would like to clarify, however, that when the manifold admits a lower bound on sectional curvature, this dependence on the footpoint can be removed, leading to a strictly sharper bound without the $d(p_0, \bar{x})$ term.
>
> Regarding the reviewer’s concern about the differentially private procedure for selecting $p_0$. In the revised manuscript, we now provide an explicit DP mechanism for estimating the footpoint. This procedure is summarized at the top of the Revision Outline and formally detailed in Appendix G of the updated PDF. We hope this resolves the concern and clarifies how $p_0$ can be selected in a fully differentially private manner.
>
> > the definition of exponential wrapped mechanism can be clearer. Is $\eta$ in definition 1 the original output $f(D)$?
>
>
> Definition 2.1 introduces the definition for the exponential-wrapped Laplace distribution, while the corresponding exponential-wrapped Laplace mechanism is formally stated in Theorem 2.1. In this context, $\eta$ is simply used to denote a location parameter for the said distribution, which we call the center. In the exponential-wrapped mechanism, we explicitly set the center $\eta$ to be $f(D)$.
>
> To improve clarity, we have revised the definitions related to the Exponential-Wrapped mechanisms. For instance, for the EWL mechanism (Theorem 2.1), we have added the line "the Exponential-Wrapped Laplace mechanism, which outputs $Y \sim \operatorname{EWL}(p_0, f(\mathcal{D}), \Delta/\varepsilon))$, satisfies $\varepsilon$-DP" where $\operatorname{EWL}(p_0, \eta, \sigma)$ is a new notation introduced for the EWL distribution with footpoint $p_0$, center $\eta$, and rate $\sigma$. See the updated PDF file for more details (changes are highlighted in blue).
>
> > Should be footpoint or footprint?
>
> It should be "footpoint". All usage of "footprint" has been changed to "footpoint" throughout the manuscript.
>
> > Can you provide a concrete DP mechanism for $p_0$ selection?
>
> A detailed description of a differentially private mechanism for selecting $p_0$ is provided in the top comment of the Revision outline. For completeness, the same mechanism is also included in Appendix G of the updated PDF file.

---

### Author Response · Authors · 2025-11-19
**Revision Outline**

We sincerely thank all reviewers for their thoughtful and constructive feedback. In response, we have revised the manuscript accordingly and uploaded an updated PDF in which all modifications are highlighted in blue. Below, we provide a summary of the corresponding changes.

## Additional Simulations

In response to Reviewer bvfn's comment on what would happen if we vary the sample size of the dataset. We have added an additional simulation in Appendix E, where we repeat the simulation for hyperbolic space but fix the privacy budget $\mu = 0.1$ while varying the sample size from $n=10$ to $n=100$. The results are provided in Figure 5. As the sample size $n$ only factors into the result via the computation of the sensitivity. The sensitivity for the sample Fr\'echet mean under Hadamard manifolds takes the form of $\Delta = 2r/n$ and the rate parameter $\sigma$ of the noises injected takes the form of $\sigma = \Delta/ \mu = 2r/(n\mu)$. Thus, one would expect $n$ and $\mu$ to have the same effect on the result. Indeed, this is what is observed in Figure 5, which mirrors what we observed in Figure 2.

In response to Reviewer G5je's comment on the simulation of how sensitive the mechanism is to the footpoint selection. We have added an additional simulation in Appendix F, where we repeat the simulation for hyperbolic space but randomly select a point within $B_{r/2}(p)$ with $p = (1, 0, \dots, 0)$ as the footpoint. The results are provided in Figure 6. Compared to Figure 2, we observed that the performance of our EWG mechanism degrades slightly but still outperforms the RL mechanism.

## Selecting a Data-Dependent Footpoint $p_0$

As pointed out by Reviewer w921 and G5je, a concrete DP mechanism for selecting a data-dependent footpoint $p_0$ would be ideal. We have added a DP mechanism for selecting $p_0$ in Appendix G. For convenience, we will state it here as well.

In the context of outputting sample fr\'echet mean, a natural candidate for the footpoint would be the extrinsic sample fr\'echet mean $\bar{x}_E$, see section 3 of [1] for more details. The extrinsic frechet mean $\bar{x}_E$ is obtained by
1. Embedding the manifold $\mathcal{M}$ into a ambient Euclidean space via the embedding $j: \mathcal{M} \to \mathbb{R}^k$: $j(X_1), \dots, j(X_n)$.
2. Compute the Euclidean mean of $j(X_1), \dots, j(X_n)$ as $\overline{j(x)}$.
3. Project the mean $\overline{j(x)}$ back to $j(\mathcal{M})$ via the projection $P$: $P(\overline{j(x)})$.
4. Lastly, map back to $\mathcal{M}$ by reverse the embedding: $\bar{x}_E = j^{-1}(P(\overline{j(x)}))$.


To obtain a differentially private version of $\bar{x}_E$, we can simply inject the Euclidean Gaussian noise into $\overline{j(x)}$ to obtain $\overline{j(x)}^{\operatorname{gdp}}$ via the Gaussian mechanism for GDP on Euclidean space. It follows that $\bar{x}_E^{\operatorname{dp}} = j^{-1}(P(\overline{j(x)}^{\operatorname{dp}}))$ is differential private by the post-processing property.

Note that this approach could extend to other $\mathcal{M}$-statistics as long as there exists a Euclidean counterpart, which is often the case in manifolds.



## Notations and Typos

As pointed out by reviewer bvfn, the assumption "$|\operatorname{Sec}_{\mathcal{M}}| < K$ with $K \ge 0$" stated in Theorem 3.1 is confusing. It has now been adjusted to "$\operatorname{Sec}_{\mathcal{M}} > K$ with $K \le 0$" instead.

As pointed out by reviewer w921 and bvfn, "footpoint" and "footprint" are used interchangeably when referring to the parameter $p_0$ of the Exponential-Wrapped distributions. We have removed any usage of "footprint" and will only use "footpoint" when referring to the parameter $p_0$.

As pointed out by reviewer w921, the definitions of Exponential-Wrapped mechanisms can be clearer. Some adjustments have been made to make it clearer. See the reply to reviewer w921 or in the updated PDF file for more details.

Sometimes $\mathcal{P}_5$ instead $S_5^+$ was used to denote the space of $5\times 5$ symmetric positive definite matrices. It has now been adjusted to $S_5^+$.

Sometimes $\hat{\eta}$ was used to denote the sample fr\'echet mean instead of $\bar{x}$. It has now been adjusted to $\bar{x}$.

Sometimes "tangent center" was used to refer to the center parameter $\eta$. It has now been adjusted to "center".

Citation errors in the first two paragraphs of the introductions have been fixed, where \\citep\{\} should be used instead of \\citet}\{\}.

---

### Author Response · Authors · 2025-12-01
**Final Comment to AC**

Dear AC,

We sincerely appreciate the reviewers’ feedback. We addressed the main concerns through theoretical clarifications, revised arguments, and additional experiments, documented in the rebuttal (Revision Outline comment) and the revised PDF (highlighted in blue).

- **Reviewer w921** finds the idea of using the exponential map on Hadamard manifolds appealing and appreciates that the paper covers multiple privacy notions, but is concerned that the Frechet mean utility bound is weak on curved manifolds due to a footpoint term $d(p_0,\bar{x})$ that does not vanish with $n$, and that the paper originally lacked a concrete DP procedure for selecting $p_0$, as well as some clarity issues in the definition of the exponential-wrapped mechanism and terminology. In our response, we explained that the $d(p_0,\bar{x})$ term arises from a triangle-inequality step and that, under a lower bound on sectional curvature, this dependence can be removed to give a sharper bound; we also introduced an explicit DP mechanism for estimating the footpoint $p_0$, clarified the mechanism definitions, and consistently use “footpoint” throughout. Reviewer w921 gives a **rating of 4 (“marginally below the acceptance threshold”) with confidence 2** and **were unable to comment further or adjust the score due to the OpenReview incident**.

- **Reviewer bvfn** finds the wrapped mechanisms and manifold/DP coverage interesting and generally well presented, but initially raised concerns about the lack of experiments varying the sample size, the formulation and readability of Theorem 3.1, various typos and inconsistent notation (notably “footPOINT”/“footPRINT”), and a small mistake in the sampling description where the second step used $\log$ instead of $\exp$. In our revision, we added new experiments in Appendix E varying $n$, clarified and simplified the conditions in Theorem 3.1, systematically fixed citation and notation issues (including consistently using “footpoint”), and corrected the sampling description to use $\exp$. Reviewer bvfn then stated that their major concerns had been addressed and **raised the score from 6 to 8** on **Nov 20th before the major information leak on Nov 27th**.

- **Reviewer Lgao** is very positive, viewing the framework for exponential-wrapped mechanisms on Hadamard manifolds as timely, mathematically interesting, and well supported by numerical experiments, and assigns a **rating of 8 (“accept, good paper (poster)”)**. The main weaknesses are that the method requires a non–data-driven footpoint and that the initial version lacked simulations on sensitivity to this choice. In response, we added simulations in Appendix F for the hyperbolic case where the footpoint is randomly selected in a smaller ball; the results show slight degradation but EWG still outperforms the RL baseline. Furthermore, we provide a data-driven footpoint selection in Appendix G.

- **Reviewer G5je** is also very positive, giving a **rating of 8 (“accept, good paper (poster)”)**, and emphasizes that the exponential-wrapped mechanisms provide an efficient, MCMC-free approach to DP on curved manifolds, are practically relevant for manifold-valued medical imaging data, and are supported by extensive experiments including the OCTMNIST dataset. The main weaknesses are that performance degrades under the affine-invariant metric in low dimensions ($d=3$) at high privacy levels due to footpoint misalignment, and that experiments focus only on mean estimation despite mentioning other tasks such as principal geodesic analysis or regression. In our response, we introduced an explicit DP mechanism for estimating the footpoint $p_0$ (detailed in Appendix G) and clarified that we deliberately focus on Frechet mean estimation as a foundational problem, with extensions to more complex tasks left for future work.

Across reviewers, the record shows a clear trend: our rebuttal and revisions addressed the main technical and conceptual concerns, particularly the Frechet mean utility bound, the choice of footpoint $p_0$, and additional experiments. Three reviewers (Lgao, G5je, and bvfn) give **scores of 8** (“accept, good paper (poster)”), and one reviewer (w921) gives a marginally-below-threshold rating of 4 with low confidence. Due to the OpenReview incident and early closure of the discussion interface, three reviewers (w921, Lgao, and G5je) did not have the opportunity to comment further or adjust their scores after our final clarifications and experiments. We hope that the AC will consider the full written record, including the resolution of concerns and the improvements made in response to reviewer feedback, when making a final decision.

---

### Meta-Review · Area_Chair_yaGQ · 2025-12-15

**Summary:**

The main concerns raised by the reviewers were the lack of a DP algorithm for selecting the footpoint $p_0$ and some concerns on clarity.

**Reviewer Concerns:**

The author rebuttal answers most concerns raised by the reviewers including revising the paper.

**Reviewer Scores:**

Most reviewers already clearly recommend acceptance, and the revision addresses the most important concerns of the single negative reviewer, so looks like a clear accept.

---

### Decision · Program_Chairs · 2026-01-26

Accept (Poster)